# DECENTRALIZED OPTIMIZATION WITH COUPLED CONSTRAINTS

**Demyan Yarmoshik**
MIPT; Research Center for Artificial Intelligence, Innopolis University, Innopolis, Russia
yarmoshik.dv@phystech.edu

**Alexander Rogozin**
MIPT; Skoltech
aleksander.rogozin@phystech.edu

**Nikita Kiselev**
Research Center for Artificial Intelligence, Innopolis University, Innopolis, Russia; MIPT
kiselev.ns@phystech.edu

**Daniil Dorin**
MIPT
dorin.dd@phystech.edu

**Alexander Gasnikov**
Research Center for Artificial Intelligence, Innopolis University, Innopolis, Russia; MIPT; Skoltech
gasnikov@yandex.ru

**Dmitry Kovalev**
Yandex Research
dakovalev1@gmail.com

## ABSTRACT

We consider the decentralized minimization of a separable objective $\sum_{i=1}^{n} f_i(x_i)$, where the variables are coupled through an affine constraint $\sum_{i=1}^{n} (\mathbf{A}_i x_i - b_i) = 0$. We assume that the functions $f_i$, matrices $\mathbf{A}_i$, and vectors $b_i$ are stored locally by the nodes of a computational network, and that the functions $f_i$ are smooth and strongly convex.

This problem has significant applications in resource allocation and systems control and can also arise in distributed machine learning. We propose lower complexity bounds for decentralized optimization problems with coupled constraints and a first-order algorithm achieving the lower bounds. To the best of our knowledge, our method is also the first linearly convergent first-order decentralized algorithm for problems with general affine coupled constraints.

## 1 INTRODUCTION

We consider the decentralized optimization problem with coupled constraints

$$\min_{x_1 \in \mathbb{R}^{d_1}, \ldots, x_n \in \mathbb{R}^{d_n}} \sum_{i=1}^{n} f_i(x_i) \quad \text{s.t.} \quad \sum_{i=1}^{n} (\mathbf{A}_i x_i - b_i) = 0, \tag{1}$$

where for $i \in \{1, \ldots, n\}$ functions $f_i(x_i) \colon \mathbb{R}^{d_i} \to \mathbb{R}$ are continuously differentiable, $\mathbf{A}_i \in \mathbb{R}^{m \times d_i}$ and $b_i \in \mathbb{R}^m$ are constraint matrices and vectors respectively.

We are interested in solving problem (1) in a decentralized distributed setting. That is, we assume the existence of a communication network $\mathcal{G} = (\mathcal{V}, \mathcal{E})$, where $\mathcal{V} = \{1, \ldots, n\}$ is the set of compute nodes, and $\mathcal{E} \subset \mathcal{V} \times \mathcal{V}$ is the set of communication links in the network. Each compute node $i \in \mathcal{V}$ locally stores the objective function $f_i(x_i)$, the constraint matrix $\mathbf{A}_i$ and the vector $b_i$. Compute node $i \in \mathcal{V}$ can send information (e.g., vectors, scalars, etc.) to compute node $j \in \mathcal{V}$ if and only if there is an edge $(i, j) \in \mathcal{E}$ in the communication network.

Coupled constraints arise in various application scenarios, where sharing resources or information takes place. Often, due to the distributed nature of such problems, decentralization is desired for communication and/or privacy related reasons. Let us briefly describe several practical cases of optimization problems with coupled constraints.

- **Optimal exchange.** Also known as the resource allocation problem Boyd et al. (2011); Nedić et al. (2018), it writes as

$$\min_{x_1, \ldots, x_n \in \mathbb{R}^d} \sum_{i=1}^{n} f_i(x_i) \quad \text{s.t.} \quad \sum_{i=1}^{n} x_i = b,$$

where $x_i \in \mathbb{R}^d$ represents the quantities of commodities exchanged among the agents of the system, and $b \in \mathbb{R}^d$ represents the shared budget or demand for each commodity. This problem is essential in economics Arrow and Debreu (1954), and systems control Dominguez-Garcia et al. (2012).

- **Problems on graphs.** In various applications, distributed systems are formed on the basis of physical networks. This is the case for electrical microgrids, telecommunication networks and drone swarms. Distributed optimization on graphs applies to such systems and encompasses, to name a few, optimal power flow Wang et al. (2016) and power system state estimation Zhang et al. (2024) problems.

As an example, consider an electric power network. Let $x_i \in \mathbb{R}^2$ denote the voltage phase angle and the magnitude at $i$-th electric node, and let $s$ be the vector of (active and reactive) power flows for each pair of adjacent electric nodes. Highly accurate linearization approaches Yang et al. (2016); Van den Bergh et al. (2014) allow to formulate the necessary relation between voltages and power flows as a linear system of equations $\sum_{i=1}^{n} \mathbf{A}_i x_i = s$. An important property of the matrices $\mathbf{A}_i$ is that their compatibility with the physical network (but not necessary with the communication network). This means that for each row of the matrix $(\mathbf{A}_1, \ldots, \mathbf{A}_n)$, there is a node $k$ such that $\mathbf{A}_i$ can have nonzero elements in this row only if nodes $i$ and $k$ are connected in the physical network, or $k = i$.

- **Consensus optimization.** Related to the previous example is the consensus optimization Boyd et al. (2011)

$$\min_{x_1, \ldots, x_n \in \mathbb{R}^d} \sum_{i=1}^{n} f_i(x_i) \quad \text{s.t.} \quad x_1 = x_2 = \ldots = x_n.$$

It is widely used in horizontal federated learning Kairouz et al. (2021), as well as in the more general context of decentralized optimization of finite-sum objectives Gorbunov et al. (2022); Scaman et al. (2017).

To handle the consensus constraint, decentralized algorithms either reformulate it as $\sum_{i=1}^{n} \mathbf{W}_i x_i = 0$, where $\mathbf{W}_i$ is the $i$-th vertical block of a gossip matrix (an example of which is the communication graph's Laplacian), or utilize the closely related mixing matrix approach Gorbunov et al. (2022). Mixing and gossip matrices are used because they are communication-friendly: calculating the sum $\sum_{i=1}^{n} \mathbf{W}_i x_i$ only requires each compute node to communicate once with each of its adjacent nodes. Clearly, consensus optimization with gossip matrix reformulation can be reduced to (1) by setting $\mathbf{A}_i = \mathbf{W}_i$. However, the principal difference between this example and (1), is that (1) does not assume $\mathbf{A}_i$ to be communication-friendly. We discuss the complexity of the reduction from consensus optimization to optimization with coupled constraints in Appendix A.

- **Vertical federated learning (VFL).** In the case of VFL, the data is partitioned by features, differing from the usual (horizontal) federated learning, where the data is partitioned by samples Yang et al. (2019); Boyd et al. (2011). Let $\mathbf{F}$ be the matrix of features, split vertically between compute nodes into submatrices $\mathbf{F}_i$, so that each node possesses its own subset of features for all data samples. Let

$l \in \mathbb{R}^m$ denote the vector of labels, and let $x_i \in \mathbb{R}^{d_i}$ be the vector of model parameters owned by the $i$-th node. VFL problem formulates as

$$\min_{\substack{z \in \mathbb{R}^m \\ x_1 \in \mathbb{R}^{d_1}, \ldots, x_n \in \mathbb{R}^{d_n}}} \ell(z, l) + \sum_{i=1}^{n} r_i(x_i) \quad \text{s.t.} \quad \sum_{i=1}^{n} \mathbf{F}_i x_i = z, \tag{2}$$

where $\ell$ is a loss function, and $r_i$ are regularizers. The constraints in (2) are coupled constraints, and the objective is separable; therefore, it is a special case of (1). We return to the VFL example in Section 6.

**Paper organization**. In Section 2 we present a literature review. Subsequently, in Section 3 we introduce the assumptions and problem parameters. Section 4 describes the key ideas of algorithm development and Section 5 presents the convergence rate of the method and the lower complexity bounds. Finally, in Section 6, we provide numerical simulations.

## 2   RELATED WORK AND OUR CONTRIBUTION

Decentralized optimization algorithms were initially proposed for consensus optimization Nedić and Ozdaglar (2009), based on earlier research in distributed optimization Tsitsiklis (1984); Bertsekas and Tsitsiklis (1989) and algorithms for decentralized averaging (*consensus* or *gossip* algorithms) Boyd et al. (2006); Olshevsky and Tsitsiklis (2009), which assumed the existence of a communication network, as does the present paper. The optimal complexity for consensus optimization was first achieved with a dual accelerated gradient descent in Scaman et al. (2017), where the method required computing gradients of Fenchel conjugates of $f_i(x)$. The corresponding complexity lower bounds were also established in the same paper. This result was later generalized to primal algorithms (which use gradients of the functions $f_i(x)$ themselves) Kovalev et al. (2020), time-varying communication graphs Li and Lin (2021); Kovalev et al. (2021) and methods that use stochastic gradients Dvinskikh and Gasnikov (2021). Today there also exist algorithms with communication compression Beznosikov et al. (2023), asynchronous algorithms Koloskova (2024), algorithms for saddle-point formulations Rogozin et al. (2021) and gradient-free oracles Beznosikov et al. (2020), making decentralized consensus optimization a quite well-developed field Nedić (2020); Gorbunov et al. (2022), benefiting systems control Ram et al. (2009) and machine learning Lian et al. (2017).

Beginning with the addition of local constraints to consensus optimization Nedic et al. (2010); Zhu and Martinez (2011), constrained decentralized optimization has been established as a research direction. A zoo of distributed problems with constraints was investigated in Necoara et al. (2011); Necoara and Nedelcu (2014; 2015).

Primarily motivated by the demand from the power systems community, various decentralized algorithms for coupled constraints have been proposed. Generally designed for versatile engineering applications, many of these algorithms assume restricted function domains Wang and Hu (2022); Liang et al. (2019); Nedić et al. (2018); Gong and Zhang (2023); Zhang et al. (2021); Wu et al. (2022), nonlinear inequality constraints Liang et al. (2019); Gong and Zhang (2023); Wu et al. (2022), time-varying graphs Zhang et al. (2021); Nedić et al. (2018) or utilize specific problem structure Wang and Hu (2022).

Works of Doan and Olshevsky (2017); Li et al. (2018); Nedić et al. (2018) focus on the resource allocation problem. For undirected time-varying graphs Doan and Olshevsky (2017) proposes a first-order algorithm with $O(B\kappa_f n^2 \ln \frac{1}{\varepsilon})$ communication and derivative computation complexity bound, where $B$ is the time required for the time-varying graph to reach connectivity. Li et al. (2018) applies a combination of gradient

Table 1: Comparison of algorithms for decentralized optimization with coupled constraints

| Reference | Oracle | Rate |
|---|---|---|
| Doan and Olshevsky (2017) [†] | First-order | Linear |
| Falsone et al. (2020) | Prox | Sub-linear |
| Wu et al. (2022) | Prox | Sub-linear |
| Chang (2016) | Prox | Sub-linear |
| Li et al. (2018) [†] | Prox | Linear |
| Gong and Zhang (2023) | Inexact prox | Linear |
| Nedić et al. (2018) [†] | First-order | Accelerated |
| This work | First-order | Optimal |

[†] Applicable only for resource allocation problem

tracking and push-sum approaches from Nedic et al. (2017) to obtain linear convergence on directed time-varying graphs in the restricted domain case, *i.e.*, $x_i \in \Omega_i$, where $\Omega_i$ is a nonempty closed convex set. Nedić et al. (2018) achieves accelerated linear convergence via a proximal point method in the restricted domain case. When $\Omega_i = \mathbb{R}^d$, they also show that Nesterov's accelerated gradient descent can be applied to achieve optimal $O(\sqrt{\kappa_{\mathbf{W}}}\sqrt{\kappa_f} \ln \frac{1}{\varepsilon})$ communication complexity. In Gong and Zhang (2023) an inexact proximal-point method is proposed to solve problems with coupled affine equality and convex inequality constraints. Linear convergence is proved when the inequalities are absent, and $\Omega_i$ are convex polyhedrons. The papers Wu et al. (2022), Chang (2016), Falsone et al. (2020) present algorithms with sub-linear convergence.

As summarized in Table 1, no accelerated linearly convergent algorithms for general affine-equality coupled constraints were present in the literature prior to our work. Also, most of the algorithms require proximal oracle, which allows to handle more general problem formulations, but has higher computational burden than the first-order oracle. We propose a new first-order decentralized algorithm with optimal (accelerated) linear convergence rate. We prove its optimality by providing lower bounds for the number of objective's gradient computations, matrix multiplications and decentralized communications, which match complexity bounds for our algorithm.

## 3 MATHEMATICAL SETTING AND ASSUMPTIONS

Let us begin by introducing the notation. The largest and smallest nonzero eigenvalues (or singular values) of a matrix $\mathbf{C}$ are denoted by $\lambda_{\max}(\mathbf{C})$ (or $\sigma_{\max}(\mathbf{C})$) and $\lambda_{\min+}(\mathbf{C})$ (or $\sigma_{\min+}(\mathbf{C})$), respectively. For vectors $x_i \in \mathbb{R}^{d_i}$ we introduce a column-stacked vector $x = \mathrm{col}(x_1, \ldots, x_m) = (x_1^\top \ldots x_m^\top)^\top \in \mathbb{R}^d$. We denote the identity matrix by $\mathbf{I}_m \in \mathbb{R}^{m \times m}$. The symbol $\otimes$ denotes the Kronecker product of matrices. By $\mathcal{L}_m$ we denote the so-called consensus space, which is given as $\mathcal{L}_m = \{(y_1, \ldots, y_n) \in (\mathbb{R}^m)^n : y_1, \ldots, y_n \in \mathbb{R}^m \text{ and } y_1 = \cdots = y_n\}$, and $\mathcal{L}_m^\perp$ denotes the orthogonal complement to $\mathcal{L}_m$, which is given as

$$\mathcal{L}_m^\perp = \{(y_1, \ldots, y_n) \in (\mathbb{R}^m)^n : y_1, \ldots, y_n \in \mathbb{R}^m \text{ and } y_1 + \cdots + y_n = 0\}. \tag{3}$$

**Assumption 1.** *Continuously differentiable functions $f_i(x) \colon \mathbb{R}^{d_i} \to \mathbb{R}$, $i \in \{1, \ldots, n\}$ are $L_f$-smooth and $\mu_f$-strongly convex, where $L_f \geq \mu_f > 0$. That is, for all $x_1, x_2 \in \mathbb{R}^{d_i}$ and $i \in \{1, \ldots, n\}$, the following inequalities hold:*

$$\frac{\mu_f}{2}\|x_2 - x_1\|^2 \leq f_i(x_2) - f_i(x_1) - \langle \nabla f_i(x_1), x_2 - x_1 \rangle \leq \frac{L_f}{2}\|x_2 - x_1\|^2.$$

*By $\kappa_f$ we denote the condition number $\kappa_f = L_f/\mu_f$.*

**Assumption 2.** *There exists $x^* = (x_1^*, \ldots, x_n^*), x_i^* \in \mathbb{R}^{d_i}$ such that $\sum_{i=1}^n (\mathbf{A}_i x_i^* - b_i) = 0$. There exist constants $L_{\mathbf{A}} \geq \mu_{\mathbf{A}} > 0$, such that the constraint matrices $\mathbf{A}_1, \ldots, \mathbf{A}_n$ satisfy the following inequalities:*

$$\sigma_{\max}^2(\mathbf{A}) = \max_{i \in \{1, \ldots, n\}} \sigma_{\max}^2(\mathbf{A}_i) \leq L_{\mathbf{A}}, \qquad \mu_{\mathbf{A}} \leq \lambda_{\min+}(\mathbf{S}), \tag{4}$$

*where the matrix $\mathbf{S} \in \mathbb{R}^{m \times m}$ is defined as $\mathbf{S} = \frac{1}{n}\sum_{i=1}^n \mathbf{A}_i \mathbf{A}_i^\top$. We also define the condition number of the block-diagonal matrix $\mathbf{A} = \mathrm{diag}(\mathbf{A}_1, \ldots, \mathbf{A}_n) \in \mathbb{R}^{mn \times d}$ as $\kappa_{\mathbf{A}} = L_{\mathbf{A}}/\mu_{\mathbf{A}}$.*

For any matrix $\mathbf{M}$ other than $\mathbf{A}$ we denote by $L_{\mathbf{M}}$ and $\mu_{\mathbf{M}}$ some upper and lower bound on its maximal and minimal positive squared singular values respectively:

$$\lambda_{\max}(\mathbf{M}^\top \mathbf{M}) = \sigma_{\max}^2(\mathbf{M}) \leq L_{\mathbf{M}}, \qquad \mu_{\mathbf{M}} \leq \sigma_{\min+}^2(\mathbf{M}) = \lambda_{\min+}(\mathbf{M}^\top \mathbf{M}). \tag{5}$$

We also assume the existence of a so-called gossip matrix $W \in \mathbb{R}^{n \times n}$ associated with the communication network $\mathcal{G}$, which satisfies the following assumption.

**Assumption 3.** *The gossip matrix $W$ is a $n \times n$ symmetric positive semidefinite matrix such that:*
  1. *$W_{ij} \neq 0$ if and only if $(i, j) \in \mathcal{E}$ or $i = j$.*
  2. *$Wy = 0$ if and only if $y \in \mathcal{L}_1$, i.e. $y_1 = \ldots = y_n$.*
  3. *There exist constants $L_{\mathbf{W}} \geq \mu_{\mathbf{W}} > 0$ such that $\mu_{\mathbf{W}} \leq \lambda_{\min+}^2(W)$ and $\lambda_{\max}^2(W) \leq L_{\mathbf{W}}$.*

We will use a dimension-lifted analogue of the gossip matrix defined as $\mathbf{W} = W \otimes \mathbf{I}_m$. From the properties of the Kronecker product of matrices it follows that $\lambda^2_{\min+}(\mathbf{W}) = \lambda^2_{\min+}(W)$ and $\lambda^2_{\max}(\mathbf{W}) = \lambda^2_{\max}(W)$. By $\kappa_{\mathbf{W}}$ we denote the condition number

$$\kappa_{\mathbf{W}} = \sqrt{\frac{L_{\mathbf{W}}}{\mu_{\mathbf{W}}}} \geq \frac{\lambda_{\max}(\mathbf{W})}{\lambda_{\min+}(\mathbf{W})}. \tag{6}$$

Moreover, the kernel and range spaces of $W$ and $\mathbf{W}$ are given by

$$\ker W = \mathcal{L}_1, \ \operatorname{range} W = \mathcal{L}_1^\perp, \quad \ker \mathbf{W} = \mathcal{L}_m, \ \operatorname{range} \mathbf{W} = \mathcal{L}_m^\perp. \tag{7}$$

## 4 DERIVATION OF THE ALGORITHM

### 4.1 STRONGLY CONVEX COMMUNICATION-FRIENDLY REFORMULATION

Let $\mathbf{W}'$ be any positive semidefinite matrix such that

$$\operatorname{range} \mathbf{W}' = (\ker \mathbf{W}')^\perp = \mathcal{L}_m^\perp, \tag{8}$$

and multiplication of a vector $y = (y_1, \dots, y_n) \in (\mathbb{R}^m)^n$ by $\mathbf{W}'$ can be performed efficiently in the decentralized manner if its $i$-th block component $y_i$ is stored at $i$-th node of the computation network. Similarly to eq. (6), we define

$$\kappa_{\mathbf{W}'} = \sqrt{\frac{L_{\mathbf{W}'}}{\mu_{\mathbf{W}'}}} \geq \frac{\lambda_{\max}(\mathbf{W}')}{\lambda_{\min+}(\mathbf{W}')}. \tag{9}$$

Due to the definition of $\mathbf{W}$ and eq. (7), the simplest choice for $\mathbf{W}'$ might be to set $\mathbf{W}' = \mathbf{W}$. Later we will specify another way to choose $\mathbf{W}'$ for optimal algorithmic performance.

Problem (1) can be reformulated as follows:

$$\min_{x \in \mathbb{R}^d, y \in (\mathbb{R}^m)^n} G(x, y) \quad \text{s.t.} \quad \mathbf{A}x + \gamma \mathbf{W}'y = \mathbf{b}, \tag{10}$$

where the function $G(x, y) \colon \mathbb{R}^d \times (\mathbb{R}^m)^n \to \mathbb{R}$ is defined as

$$G(x, y) = F(x) + \frac{r}{2}\|\mathbf{A}x + \gamma \mathbf{W}'y - \mathbf{b}\|^2, \tag{11}$$

the function $F(x) \colon \mathbb{R}^d \to \mathbb{R}$ is defined as $F(x) = \sum_{i=1}^n f_i(x_i)$, where $x = (x_1, \dots, x_n), x_i \in \mathbb{R}^{d_i}$, the matrix $\mathbf{A} \in \mathbb{R}^{mn \times d}$ is the block-diagonal matrix $\mathbf{A} = \operatorname{diag}(\mathbf{A}_1, \dots, \mathbf{A}_n)$, the vector $\mathbf{b}$ is the column-stacked vector $\mathbf{b} = \operatorname{col}(b_1, \dots, b_n) \in \mathbb{R}^{mn}$, and $r, \gamma > 0$ are scalar constants that will be determined later.

From the definitions of $\mathbf{A}$, $\mathbf{b}$ and $\mathcal{L}_m^\perp$ (eq. (3)) it is clear that $\sum_{i=1}^n (\mathbf{A}_i x_i - b_i) = 0$ if and only if $\mathbf{A}x - \mathbf{b} \in \mathcal{L}_m^\perp$. Since $\operatorname{range} \mathbf{W}' = \mathcal{L}_m^\perp$, the constraint in problem (10) is equivalent to the coupled constraint in (1). For all $x, y$ satisfying the constraint, the augmented objective function $G(x, y)$ is equal to the original objective function $F(x)$. Therefore, problem (10) is equivalent to problem (1).

The following Lemma 1 shows that the function $G(x, y)$ is strongly convex and smooth.

**Lemma 1.** *Let $r$ and $\gamma$ be defined as follows:*

$$r = \frac{\mu_f}{2L_{\mathbf{A}}}, \quad \gamma^2 = \frac{\mu_{\mathbf{A}} + L_{\mathbf{A}}}{\mu_{\mathbf{W}'}}. \tag{12}$$

*Then, the strong convexity and smoothness constants of $G(x, y)$ on $\mathbb{R}^d \times \mathcal{L}_m^\perp$ are given by*

$$\mu_G = \mu_f \min\left\{\frac{1}{2}, \frac{\mu_{\mathbf{A}} + L_{\mathbf{A}}}{4L_{\mathbf{A}}}\right\}, \quad L_G = \max\left\{L_f + \mu_f, \mu_f \frac{\mu_{\mathbf{A}} + L_{\mathbf{A}}}{L_{\mathbf{A}}} \frac{L_{\mathbf{W}'}}{\mu_{\mathbf{W}'}}\right\}. \tag{13}$$

Let the matrix $\mathbf{B} \in \mathbb{R}^{mn \times (d+mn)}$ be defined as $\mathbf{B} = [\mathbf{A} \quad \gamma \mathbf{W}']$. The following Lemma 2 connects the spectral properties of $\mathbf{B}$, $\mathbf{A}$ and $\mathbf{W}'$.

**Lemma 2.** *The following bounds on the singular values of* $\mathbf{B}$ *hold:*

$$\sigma^2_{\min+}(\mathbf{B}) \geq \mu_{\mathbf{B}} = \frac{\mu_{\mathbf{A}}}{2}, \quad \sigma^2_{\max}(\mathbf{B}) \leq L_{\mathbf{B}} = L_{\mathbf{A}} + (L_{\mathbf{A}} + \mu_{\mathbf{A}})\frac{L_{\mathbf{W}'}}{\mu_{\mathbf{W}'}}, \tag{14}$$

*and*

$$\frac{\sigma^2_{\max}(\mathbf{B})}{\sigma^2_{\min+}(\mathbf{B})} \leq \frac{L_{\mathbf{B}}}{\mu_{\mathbf{B}}} = \kappa_{\mathbf{B}} = 2\left(\kappa_{\mathbf{A}} + \frac{L_{\mathbf{W}'}}{\mu_{\mathbf{W}'}}(1 + \kappa_{\mathbf{A}})\right). \tag{15}$$

Proofs of Lemma 1 and Lemma 2 are provided in Appendix B.

## 4.2 CHEBYSHEV ACCELERATION

Chebyshev acceleration allows us to decouple the number of computations of the objective's gradient $\nabla F(x)$ from the properties of the communication network and the constraint matrix — specifically, from the condition numbers $\kappa_{\mathbf{W}}$ and $\kappa_{\mathbf{A}}$. The Chebyshev trick enables to replace the matrix with a matrix polynomial with a better condition number.

Consider some affine relation $\mathbf{M}u = \mathbf{d}$ and let $\mathcal{P}_{\mathbf{M}}$ be a polynomial such that $\mathcal{P}_{\mathbf{M}}(\lambda) = 0 \Leftrightarrow \lambda = 0$ for any eigenvalue $\lambda$ of $\mathbf{M}^\top \mathbf{M}$. Note that here we interchangeably use $\mathcal{P}$ as a polynomial of a matrix and a polynomial of a scalar. We denote any feasible point for the constraint $\mathbf{M}u = \mathbf{d}$ as $u_0$. Then,

$$\mathbf{M}u = \mathbf{d} \Leftrightarrow \mathbf{M}(u - u_0) = 0 \overset{(a)}{\Leftrightarrow} \mathbf{M}^\top \mathbf{M}(u - u_0) = 0$$
$$\overset{(b)}{\Leftrightarrow} \mathcal{P}_{\mathbf{M}}(\mathbf{M}^\top \mathbf{M})(u - u_0) = 0 \overset{(c)}{\Leftrightarrow} \sqrt{\mathcal{P}_{\mathbf{M}}(\mathbf{M}^\top \mathbf{M})}(u - u_0) = 0$$

where (a) and (c) is due to $\ker \mathbf{M}^\top \mathbf{M} = \ker \mathbf{M}$; (b) is due to $\ker \mathcal{P}_{\mathbf{M}}(\mathbf{M}^\top \mathbf{M}) = \ker \mathbf{M}^\top \mathbf{M}$ by the assumption about $\mathcal{P}_{\mathbf{M}}(\lambda)$.

Following Salim et al. (2022a) and Scaman et al. (2017), we use the translated and scaled Chebyshev polynomials, because they are the best at compressing the spectrum Auzinger and Melenk (2011).

**Lemma 3** (Salim et al. (2022a), Section 6.3.2). *Consider a matrix* $\mathbf{M}$. *Let* $\ell = \left\lceil \sqrt{\frac{L_{\mathbf{M}}}{\mu_{\mathbf{M}}}} \right\rceil \geq \left\lceil \sqrt{\frac{\lambda_{\max}(\mathbf{M}^\top \mathbf{M})}{\lambda_{\min+}(\mathbf{M}^\top \mathbf{M})}} \right\rceil$. *Define* $\mathcal{P}_{\mathbf{M}}(t) = 1 - \frac{T_\ell((L_{\mathbf{M}}+\mu_{\mathbf{M}}-2t)/(L_{\mathbf{M}}-\mu_{\mathbf{M}}))}{T_\ell((L_{\mathbf{M}}+\mu_{\mathbf{M}})/(L_{\mathbf{M}}-\mu_{\mathbf{M}}))}$, *where* $T_\ell$ *is the Chebyshev polynomial of the first kind of degree* $n$ *defined by* $T_\ell(t) = \frac{1}{2}\left(\left(t + \sqrt{t^2-1}\right)^\ell + \left(t - \sqrt{t^2-1}\right)^\ell\right)$. *Then,* $\mathcal{P}_{\mathbf{M}}(0) = 0$, *and*

$$\lambda_{\max}\left(\mathcal{P}_{\mathbf{M}}(\mathbf{M}^\top \mathbf{M})\right) \leq \max_{t \in [\mu_{\mathbf{M}}, L_{\mathbf{M}}]} \mathcal{P}_{\mathbf{M}}(t) \leq \frac{19}{15}, \tag{16}$$

$$\lambda_{\min+}\left(\mathcal{P}_{\mathbf{M}}(\mathbf{M}^\top \mathbf{M})\right) \geq \min_{t \in [\mu_{\mathbf{M}}, L_{\mathbf{M}}]} \mathcal{P}_{\mathbf{M}}(t) \geq \frac{11}{15}. \tag{17}$$

Results of this section are summarized in the following Lemma 4.

**Lemma 4.** *Define*

$$\mathbf{W}' = \mathcal{P}_{\sqrt{\overline{\mathbf{W}}}}(\mathbf{W}) \tag{18}$$

*and*

$$\mathbf{K} = \sqrt{\mathcal{P}_{\mathbf{B}}(\mathbf{B}^\top \mathbf{B})}. \tag{19}$$

*Let* $G(u) = G(x, y)$, $\mathcal{U} = \mathbb{R}^d \times \mathcal{L}_m^\perp$ *and* $\mathbf{b}' = \sqrt{\mathcal{P}_{\mathbf{B}}(\mathbf{B}^\top \mathbf{B})}u_0$. *Then, problem*

$$\min_{u \in \mathcal{U}} G(u) \quad s.t. \quad \mathbf{K}u = \mathbf{b}' \tag{20}$$

*is an equivalent preconditioned reformulation of problem* (10)*, and, in turn, of problem* (1)*.*

## 4.3 BASE ALGORITHM

Our base algorithm, Algorithm 1, is the Proximal Alternating Predictor-Corrector (PAPC) with Nesterov's acceleration, called Accelerated PAPC (APAPC). It was proposed in Salim et al. (2022a) to obtain an optimal algorithm for optimization problems formulated as (20). See Kovalev et al. (2020); Salim et al. (2022b) for the review of related algorithms and history of their development.

---

**Algorithm 1** APAPC

1: **Parameters:** $u^0 \in \mathcal{U}$ $\eta, \theta, \alpha > 0, \tau \in (0,1)$
2: Set $u_f^0 = u^0$, $z^0 = 0 \in \mathcal{U}$
3: **for** $k = 0, 1, 2, \ldots$ **do**
4: $\quad u_g^k := \tau u^k + (1-\tau) u_f^k$
5: $\quad u^{k+\frac{1}{2}} := (1 + \eta\alpha)^{-1}(u^k - \eta(\nabla G(u_g^k) - \alpha u_g^k + z^k))$
6: $\quad z^{k+1} := z^k + \theta \mathbf{K}^\top(\mathbf{K} u^{k+\frac{1}{2}} - \mathbf{b}')$
7: $\quad u^{k+1} := (1 + \eta\alpha)^{-1}(u^k - \eta(\nabla G(u_g^k) - \alpha u_g^k + z^{k+1}))$
8: $\quad u_f^{k+1} := u_g^k + \frac{2\tau}{2-\tau}(u^{k+1} - u^k)$
9: **end for**

---

APAPC algorithm formulates as Algorithm 1, and its convergence properties are given in Proposition 1.

**Proposition 1** (Salim et al. (2022a), Proposition 1). *Assume that the matrix $\mathbf{K}$ in (20) satisfies $\mu_\mathbf{K} > 0$ and $\mathbf{b}' \in \operatorname{range} \mathbf{K}$, and denote $\kappa_\mathbf{K} = \frac{L_\mathbf{K}}{\mu_\mathbf{K}}$. Also assume that the function $G$ is $L_G$-smooth and $\mu_G$-strongly convex. Set the parameter values of Algorithm 1 as $\tau = \min\left\{1, \frac{1}{2}\sqrt{\frac{\kappa_\mathbf{K}}{\kappa_G}}\right\}$, $\eta = \frac{1}{4\tau L_G}$, $\theta = \frac{1}{\eta L_\mathbf{K}}$ and $\alpha = \mu_G$. Denote by $u^*$ the solution of problem (20) and by $z^*$ the solution of its dual problem satisfying $z^* \in \operatorname{range} \mathbf{K}$. Then the iterates $u^k, z^k$ of Algorithm 1 satisfy*

$$\frac{1}{\eta}\left\|u^k - u^\star\right\|^2 + \frac{\eta\alpha}{\theta(1 + \eta\alpha)}\left\|(\mathbf{K}^\top)^\dagger z^k - z^\star\right\|^2 \tag{21}$$

$$+ \frac{2(1-\tau)}{\tau}\mathrm{D}_G(u_f^k, u^\star) \leq \left(1 + \frac{1}{4}\min\left\{\frac{1}{\sqrt{\kappa_G \kappa_\mathbf{K}}}, \frac{1}{\kappa_\mathbf{K}}\right\}\right)^{-k} C,$$

*where $C := \frac{1}{\eta}\left\|u^0 - u^\star\right\|^2 + \frac{1}{\theta}\|z^0 - z^\star\|^2 + \frac{2(1-\tau)}{\tau}\mathrm{D}_G(u_f^0, u^\star)$, and $\mathrm{D}_G$ denotes the Bregman divergence of $G$, defined by $D_G(u', u) = G(u') - G(u) - \langle\nabla G(u), u' - u\rangle$.*

# 5 MAIN RESULTS

## 5.1 ALGORITHM

As stated in Lemma 4, problem (20) is equivalent to problem (1). Due to Lemma 1, its objective is strongly convex, allowing us to apply Algorithm 1 to it. Using Lemma 3, we obtain that the condition numbers of $\mathbf{W}'$ and $\mathbf{K}$ are bounded as $O(1)$, but a single multiplication by $\mathbf{W}'$ and $\mathbf{K}, \mathbf{K}^\top$ translates to $O(\sqrt{\kappa_\mathbf{W}})$ multiplications by $\mathbf{W}$ and $O(\sqrt{\kappa_\mathbf{B}})$ multiplications by $\mathbf{B}, \mathbf{B}^\top$ respectively.

We implement multiplications by $\mathbf{W}'$ and $\mathbf{K}, \mathbf{K}^\top$ through numerically stable Chebyshev iteration procedures given in Algorithms 3 and 5, which only use decentralized communications and multiplications

---

**Algorithm 2** Main algorithm

1: **Parameters:** $x^0 \in \mathbb{R}^d$ $\eta, \theta, \alpha > 0, \tau \in (0,1)$
2: Set $y^0 := 0 \in (\mathbb{R}^m)^n$, $u^0 := (x^0, y^0)$,
3: $\quad u_f^0 := u^0, z^0 := 0 \in \mathbb{R}^d \times (\mathbb{R}^m)^n$
4: **for** $k = 0, 1, 2, \ldots$ **do**
5: $\quad u_g^k := \tau u^k + (1-\tau) u_f^k$
6: $\quad g^k := \mathbf{grad\_G}(u_g^k) - \alpha u_g^k$
7: $\quad u^{k+\frac{1}{2}} := (1 + \eta\alpha)^{-1}(u^k - \eta(g^k + z^k))$
8: $\quad z^{k+1} := z^k + \theta \cdot \mathbf{K\_Chebyshev}(u^{k+\frac{1}{2}})$
9: $\quad u^{k+1} := (1 + \eta\alpha)^{-1}(u^k - \eta(g^k + z^{k+1}))$
10: $\quad u_f^{k+1} := u_g^k + \frac{2\tau}{2-\tau}(u^{k+1} - u^k)$
11: **end for**

---

by $\mathbf{A}, \mathbf{A}^\top$. Lemmas 1 to 3 allow us to express the complexity of Algorithm 1 in terms of the parameters of the initial problem given in Assumptions 1 to 3. All this leads us to the following Theorem 1, a detailed proof of which is provided in Appendix B.3, as well as the derivation of Algorithms 3 and 5 and values of the parameters of Algorithm 2.

**Theorem 1.** *Set the parameter values of Algorithm 2 as $\tau = \min\left\{1, \frac{1}{2}\sqrt{\frac{19}{44 \max\{1 + \kappa_f, 6\}}}\right\}$, $\eta = \frac{1}{4\tau \max\{L_f + \mu_f, 6\mu_f\}}$, $\theta = \frac{15}{19\eta}$ and $\alpha = \frac{\mu_f}{4}$. Denote by $x^*$ the solution of problem (1).*

**Algorithm 3 mulW$'(y)$: Multiplication by W$'$**

1: **Parameters:** $y$
2: $\rho := \left(\sqrt{L_{\mathbf{W}}} - \sqrt{\mu_{\mathbf{W}}}\right)^2 /16$
3: $\nu := \left(\sqrt{L_{\mathbf{W}}} + \sqrt{\mu_{\mathbf{W}}}\right)/2$
4: $\delta^0 := -\nu/2, \ n := \lceil \sqrt{\kappa_{\mathbf{W}}} \rceil$
5: $p^0 := -\mathbf{W}y/\nu, \ y^1 := y + p^0$
6: **for** $i = 1, \ldots, n-1$ **do**
7: $\quad \beta^{i-1} := \rho/\delta^{i-1}$
8: $\quad \delta^i := -(\nu + \beta^{i-1})$
9: $\quad p^i := \left(\mathbf{W}y^i + \beta^{i-1}p^{i-1}\right)/\delta^i$
10: $\quad y^{i+1} := y^i + p^i$
11: **end for**
12: **Output:** $y - y^n$

---

**Algorithm 4 grad_G$(u)$: Computation of $\nabla G(u)$**

1: **Parameters:** $u = (x, y)$
2: $z := r\left(\mathbf{A}x + \gamma \cdot \mathbf{mulW}'(y) - \mathbf{b}\right)$
3: **Output:** $\begin{pmatrix} \nabla F(x) + \mathbf{A}^\top z \\ \gamma \cdot \mathbf{mulW}'(z) \end{pmatrix}$

---

**Algorithm 5 K_Chebyshev$(u)$: Computation of $\mathbf{K}^\top(\mathbf{K}u - \mathbf{b}')$**

1: **Parameters:** $u = (x, y)$
2: $\rho := \left(L_{\mathbf{B}} - \mu_{\mathbf{B}}\right)^2 /16$
3: $\nu := \left(L_{\mathbf{B}} + \mu_{\mathbf{B}}\right)/2$
4: $\delta^0 := -\nu/2, \ n := \lceil \sqrt{\kappa_{\mathbf{B}}} \rceil$
5: $q^0 := \mathbf{A}x + \gamma \cdot \mathbf{mulW}'(y) - \mathbf{b}$
6: $p^0 := -\dfrac{1}{\nu}\begin{pmatrix} \mathbf{A}^\top q^0 \\ \gamma \cdot \mathbf{mulW}'(q^0) \end{pmatrix}$
7: $u^1 := u + p^0$
8: **for** $i = 1, \ldots, n-1$ **do**
9: $\quad \beta^{i-1} := \rho/\delta^{i-1}$
10: $\quad \delta^i := -(\nu + \beta^{i-1})$
11: $\quad (x^i, y^i) = u^i$
12: $\quad q^i := \mathbf{A}x^i + \gamma \cdot \mathbf{mulW}'(y^i) - \mathbf{b}$
13: $\quad p^i := \dfrac{1}{\delta^i}\begin{pmatrix} \mathbf{A}^\top q^i \\ \gamma \cdot \mathbf{mulW}'(q^i) \end{pmatrix} + \beta^{i-1}p^{i-1}/\delta^i$
14: $\quad u^{i+1} := u^i + p^i$
15: **end for**
16: **Output:** $u - u^n$

---

Then, for every $\varepsilon > 0$, Algorithm 2 finds $x^k$ for which $\|x^k - x^*\|^2 \leq \varepsilon$ using $O(\sqrt{\kappa_f} \log(1/\varepsilon))$ objective's gradient computations, $O(\sqrt{\kappa_f}\sqrt{\kappa_{\mathbf{A}}} \log(1/\varepsilon))$ multiplications by $\mathbf{A}$ and $\mathbf{A}^\top$, and $O(\sqrt{\kappa_f}\sqrt{\kappa_{\mathbf{A}}}\sqrt{\kappa_{\mathbf{W}}} \log(1/\varepsilon))$ communication rounds (multiplications by $\mathbf{W}$).

## 5.2 LOWER BOUNDS

Let us formulate the lower complexity bounds for decentralized optimization with affine constraints. To do that, we formalize the class of the algorithms of interest. In the literature, approaches with continuous time Scaman et al. (2017) and discrete time Kovalev et al. (2021) are used. We use the latter discrete time formalization. We assume that the method works in synchronized rounds of three types: local objective's gradient computations, local matrix multiplications and communications. At each time step, algorithm chooses one of the three step types.

Since the devices may have different dimensions $d_i$ of locally held vectors $x_i$, they cannot communicate these vectors directly. Instead, the nodes exchange quantities $\mathbf{A}_i x_i \in \mathbb{R}^m$. For this reason, we introduce two types of memory $\mathcal{M}_i(k)$ and $\mathcal{H}_i(k)$ for node $i$ at step $k$. Set $\mathcal{M}_i(k)$ stands for the local memory that the node does not share and $\mathcal{H}_i(k)$ denotes the memory that the node exchanges with neighbors. The interaction between $\mathcal{M}_i(k)$ and $\mathcal{H}_i(k)$ is performed via multiplications by $\mathbf{A}_i$ and $\mathbf{A}_i^\top$.

Memory is initialized as $\mathcal{M}_i(0) = \{0\}$, $\mathcal{H}_i(0) = \{0\}$. Below we describe how the sets $\mathcal{M}_i(k), \mathcal{H}_i(k)$ are updated.

1. Algorithm performs local gradient comutation round at step $k$. Gradient updates only operate in $\mathcal{M}_i(k)$ and do not affect $\mathcal{H}_i(k)$. For all $i \in \mathcal{V}$ we have

$$\mathcal{M}_i(k+1) = \text{Span}\left\{x, \nabla f_i(x), \nabla f_i^*(x) : \ x \in M_i(k)\right\}, \ \mathcal{H}_i(k+1) = \mathcal{H}_i(k),$$

where $f_i^*$ is the Fenchel conjugate of $f_i$.
2. Algorithm performs local matrix multiplication round at step $k$. Sets $\mathcal{H}_i(k)$ and $\mathcal{M}_i(k)$ make mutual updates via multiplication by $\mathbf{A}_i$ and $\mathbf{A}_i^\top$. For all $i \in \mathcal{V}$ we have

$$\mathcal{M}_i(k+1) = \text{Span}\left\{\mathbf{A}_i^\top b_i, \ \mathbf{A}_i^\top y : \ y \in \mathcal{H}_i(k)\right\}, \ \mathcal{H}_i(k+1) = \text{Span}\left\{b_i, \mathbf{A}_i x : \ x \in \mathcal{M}_i(k)\right\}.$$

3. Algorithm performs a communication round at step $k$. The non-shared local memory $\mathcal{M}_i(k)$ stays unchanged, while the shared memory $\mathcal{H}_i(k+1)$ is updated via interaction with neighbors. For all

$i \in \mathcal{V}$ we have

$$\mathcal{M}_i(k+1) = \mathcal{M}_i(k), \; \mathcal{H}_i(k+1) = \text{Span} \left\{ \mathcal{H}_j(k) : \; (i,j) \in \mathcal{E} \right\}.$$

Under given memory and computation model, we formulate the lower complexity bounds.

**Theorem 2.** *For any $L_f > \mu_f > 0$, $\kappa_{\mathbf{A}}, \kappa_{\mathbf{W}} > 0$ there exist $L_f$-smooth $\mu_f$-strongly convex functions $\{f_i\}_{i=1}^n$, matrices $\mathbf{A}_i$ such that $\kappa_{\mathbf{A}} = L_{\mathbf{A}} / \mu_{\mathbf{A}}$ (where $L_{\mathbf{A}}, \mu_{\mathbf{A}}$ are defined in (4)), and a communication graph $\mathcal{G}$ with a corresponding gossip matrix $\mathbf{W}$ such that $\kappa_{\mathbf{W}} = \lambda_{\max}(\mathbf{W}) / \lambda_{\min}^+(\mathbf{W})$, for which any first-order decentralized algorithm on problem (1) to reach accuracy $\varepsilon$ requires at least*

$$N_f = \Omega\left( \sqrt{\kappa_f} \log\left(\frac{1}{\varepsilon}\right) \right) \text{ gradient computations,}$$

$$N_{\mathbf{A}} = \Omega\left( \sqrt{\kappa_f} \sqrt{\kappa_{\mathbf{A}}} \log\left(\frac{1}{\varepsilon}\right) \right) \text{ multiplications by } \mathbf{A} \text{ and } \mathbf{A}^\top,$$

$$N_{\mathbf{W}} = \Omega\left( \sqrt{\kappa_f} \sqrt{\kappa_{\mathbf{A}}} \sqrt{\kappa_{\mathbf{W}}} \log\left(\frac{1}{\varepsilon}\right) \right) \text{ communication rounds (multiplications by } \mathbf{W}).$$

A proof of Theorem 2 is provided in Appendix C.

## 6 EXPERIMENTS

The experiments were run on CPU Intel(R) Core(TM) i9-7980XE, with 62.5 GB RAM.

• **Synthetic linear regression.** In this section we perform numerical experiments on a synthetic linear regression problem with $\ell_2$-regularization:

$$\min_{x_1,\dots,x_n \in \mathbb{R}^{d_i}} \sum_{i=1}^n \left( \frac{1}{2} \| C_i x_i - d_i \|_2^2 + \frac{\theta}{2} \| x_i \|_2^2 \right) \quad \text{s.t.} \quad \sum_{i=1}^n (\mathbf{A}_i x_i - b_i) = 0, \quad (22)$$

where we randomly generate matrices $C_i \in \mathbb{R}^{d_i \times d_i}$, $\mathbf{A}_i \in \mathbb{R}^{m \times d_i}$ and vectors $d_i \in \mathbb{R}^{d_i}$, $b_i \in \mathbb{R}^m$ from the standard normal distribution. Local variables $x_i \in \mathbb{R}^{d_i}$ have the same dimension $d_i$, equal for all devices. Regularization parameter $\theta$ is $10^{-3}$. In the Fig. 1 we demonstrate the performance of the our method on the problem, that has the following parameters: $\kappa_f = 3140$, $\kappa_{\mathbf{A}} = 27$, $\kappa_{\mathbf{W}} = 89$. There we use Erdős–Rényi graph topology with $n = 20$ nodes. Local variables dimension is $d_i = 3$ and number of linear constraints is $m = 10$. We compare performance of Algorithm 2 with Tracking-ADMM algorithm Falsone et al. (2020) and DPMM algorithm Gong and Zhang (2023). Note that Tracking-ADMM and DPMM are proximal algorithms that solve a subproblem at each iteration. The choice of objective function in our simulations (linear regression) makes the corresponding proximal operator effectively computable via Conjugate Gradient algorithm Nesterov (2004) that uses gradient computations. Therefore, we measure the computational complexity of these methods in the number of gradient computations, not the number of proximal operator computations.

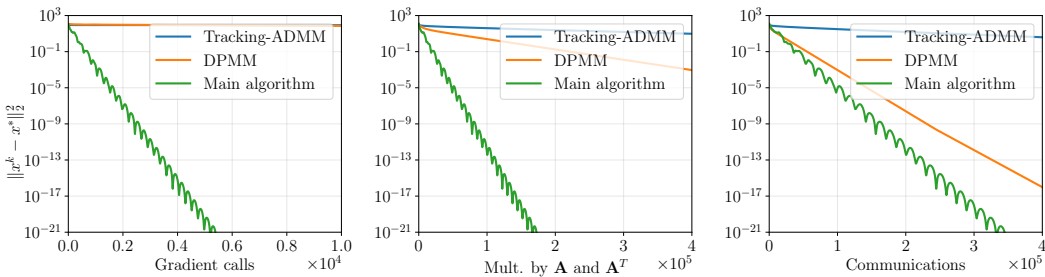

Figure 1: Synthetic, Erdős–Rényi graph, $n = 20$, $d_i = 3$, $m = 10$

• **VFL linear regression on real data.** Now we return to the problem, that we have announced in the introduction section. We apply VFL in the linear regression problem: $\ell$ is a typical mean squared loss

function, that is $\ell(z, l) = \frac{1}{2}\|z - l\|_2^2$, and $r_i$ are $\ell_2$-regularizers, *i.e.* $r_i(x_i) = \lambda\|x_i\|_2^2$. To adapt this from (2) to (1), we redefine $x_1 := \binom{x_1}{z}$ and $x_2 := x_2, \ldots, x_n := x_n$. Thus, we can derive constraints matrices as in the (1):

$$\mathbf{A}_1 = (\mathbf{F}_1 \quad -\mathbf{I}), \qquad \mathbf{A}_1 x_1 = \mathbf{F}_1 w_1 - z, \tag{23}$$

$$\mathbf{A}_i = \mathbf{F}_i, \quad i = 2, \ldots, n, \qquad \sum_{i=1}^{n} \mathbf{A}_i x_i = \sum_{i=1}^{n} \mathbf{F}_i w_i - z. \tag{24}$$

For numerical simulation, we use `mushrooms` dataset from LibSVM library Chang and Lin (2011). We split $m = 100$ samples subset vertically between $n = 7$ devices. Regularization parameter $\lambda = 10^{-2}$. The results are in the Fig. 2.

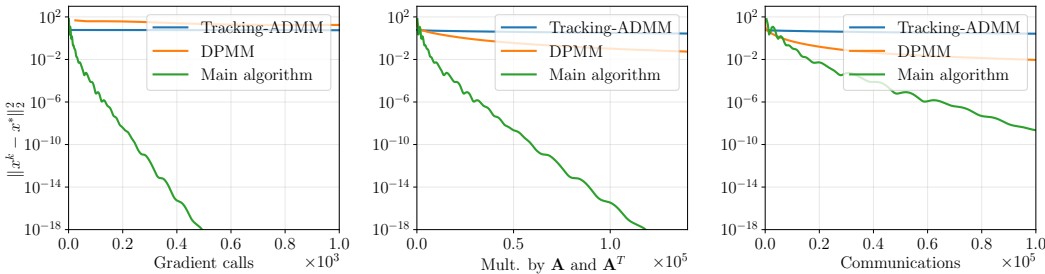

Figure 2: VFL, Erdős–Rényi graph, $n = 7$, $m = 100$

Our algorithm exhibits the best convergence rates, as evidenced by the steepest slopes. The slopes vary for gradient calls, matrix multiplications, and communications. This is due to the fact that Algorithm 2 involves many communications per iteration, in contrast to DPMM and Tracking-ADMM, which make numerous gradient calls per iteration.

## 7 ACKNOWLEDGEMENTS

The work was supported by MIPT based Center of National Technology Initiatives in the field of Artificial Intelligence for the purposes of the "road map" of Artificial Intelligence development up to 2030 and supported by NTI Foundation (agreement No.70-2021-00207 dated 22.11.2021, identifier 000000S507521QYL0002).

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

## APPENDIX / SUPPLEMENTAL MATERIAL

## A  REDUCTION OF CONSENSUS OPTIMIZATION TO COUPLED CONSTRAINTS

As mentioned in Section 1, the consensus optimization problem

$$\min_{x_1,\ldots,x_n \in \mathbb{R}^d} \sum_{i=1}^{n} f_i(x_i) \quad \text{s.t.} \quad x_1 = x_2 = \ldots = x_n$$

can be reduced to the problem with coupled constraints (1). During the review process, we were asked whether the complexity of Algorithm 2 could be reduced to match that of optimal algorithms for consensus optimization Scaman et al. (2017).

It turns out that, for a general-purpose first-order decentralized algorithm for problems with coupled constraints (as defined in Section 5.2), the communication complexity is worse by at least a factor of $\sqrt{n}$. This conclusion is based on our complexity lower bounds in Theorem 2 and the following lower bound on $\kappa_{\mathbf{A}}$ for any suitable matrix $\mathbf{A}$.

Let $d = 1$. Consider an arbitrary horizontal-block matrix $\mathbf{A}' = (\mathbf{A}_1 \ldots \mathbf{A}_n) \in \mathbb{R}^{m \times n}$ such that $\ker \mathbf{A}' = \mathcal{L}_1 = \mathrm{Span}\{\vec{1}_n\}$, which corresponds to the consensus constraint. By definition (4), we have

$$\mu_{\mathbf{A}} = \frac{1}{n}\lambda_{\min^+}\left(\sum_{i=1}^{n} \mathbf{A}_i \mathbf{A}_i^\top\right) = \frac{1}{n}\lambda_{\min^+}\left(\mathbf{A}' \mathbf{A}'^\top\right) = \frac{1}{n}\sigma_{\min^+}^2(\mathbf{A}'),$$

and

$$L_{\mathbf{A}} = \max_{i=1\ldots n} \sigma_{\max}^2(\mathbf{A}_i) = \|\mathbf{A}_{i^*}\|^2,$$

where $i^*$ denotes the index of the column achieving the maximum. We can upper bound the minimal positive singular value using a vector $v$, whose components are all zero except for one $1$ and one $-1$, implying $\|v\|^2 = 2$. Since the sum of its components is zero, $v$ is orthogonal to $\ker \mathbf{A}'$, thus

$$\sigma_{\min^+}^2(\mathbf{A}') = \min_{\substack{v:\ \|v\|>0, \\ v \perp \vec{1}}} \|\mathbf{A}'v\|^2/\|v\|^2 \le \|\mathbf{A}_i - \mathbf{A}_j\|^2/2 \le 2\|\mathbf{A}_{i^*}\|^2.$$

It follows that

$$\kappa_{\mathbf{A}} = \frac{L_{\mathbf{A}}}{\mu_{\mathbf{A}}} \ge \frac{\|\mathbf{A}_{i^*}\|^2}{\frac{2}{n}\|\mathbf{A}_{i^*}\|^2} = \frac{n}{2}.$$

This bound is nearly tight, because we can achieve $\kappa_{\mathbf{A}} = n - 1$ by setting $\mathbf{A}'$ to be the Laplacian matrix or the incidence matrix of a complete graph on $n$ nodes.

Substituting this into the complexity lower bounds in Theorem 2, we find that the number of communication rounds is lower bounded by $\Omega\left(\sqrt{\kappa_f}\sqrt{n}\sqrt{\kappa_{\mathbf{W}}}\log\left(\frac{1}{\varepsilon}\right)\right)$ for any choice of $\mathbf{A}$, while the optimal complexity for first order decentralized consensus optimization is $O\left(\sqrt{\kappa_f}\sqrt{\kappa_{\mathbf{W}}}\log\left(\frac{1}{\varepsilon}\right)\right)$.

## B  MISSING PROOFS FROM SECTION 4

### B.1  PROOF OF LEMMA 1

*Proof.* Let $D_G(x', y'; x, y)$ denote the Bregman divergence of $G$:

$$D_G(x', y'; x, y) = G(x', y') - G(x, y) - \langle \nabla_x G(x, y), x' - x \rangle - \langle \nabla_y G(x, y), y' - y \rangle. \quad (25)$$

The value of $\mu_G$ can be obtained as follows:

$$\mathrm{D}_G(x', y'; x, y) = \mathrm{D}_F(x'; x) + \frac{r}{2}\|\mathbf{A}(x' - x) + \gamma\mathbf{W}'(y' - y)\|^2$$

$$\overset{(a)}{\geq} \frac{\mu_f}{2}\|x' - x\|^2 + \frac{r}{2}\|\mathbf{A}(x' - x) + \gamma\mathbf{W}'(y' - y)\|^2$$

$$= \frac{\mu_f}{2}\|x' - x\|^2 + \frac{r}{2}\|\mathbf{A}(x' - x)\|^2 + r\langle\mathbf{A}(x' - x), \gamma\mathbf{W}'(y' - y)\rangle$$

$$+ \frac{r}{2}\|\gamma\mathbf{W}'(y' - y)\|^2$$

$$\overset{(b)}{\geq} \frac{\mu_f}{2}\|x' - x\|^2 + \frac{r}{4}\|\gamma\mathbf{W}'(y' - y)\|^2 - \frac{r}{2}\|\mathbf{A}(x' - x)\|^2$$

$$\overset{(c)}{\geq} \frac{\mu_f}{2}\|x' - x\|^2 + \frac{r\gamma^2\mu_{\mathbf{W}'}}{4}\|y' - y\|^2 - \frac{rL_{\mathbf{A}}}{2}\|x' - x\|^2$$

$$\overset{(d)}{=} \frac{\mu_f}{4}\|x' - x\|^2 + \frac{\mu_f\gamma^2\mu_{\mathbf{W}'}}{8L_{\mathbf{A}}}\|y' - y\|^2,$$

$$\overset{(e)}{\geq} \frac{\mu_f}{2}\min\left\{\frac{1}{2}, \frac{\mu_{\mathbf{A}} + L_{\mathbf{A}}}{4L_{\mathbf{A}}}\right\}\left\|\begin{pmatrix}x' - x \\ y' - y\end{pmatrix}\right\|^2,$$

where (a) is due to Assumption 1; (b) is due to Young's inequality; (c) is due to Assumption 2, $y' - y \in \mathcal{L}_m^\perp$, eq. (8) and eq. (5); (d) and (e) is due to eq. (12).

The value of $L_G$ can be obtained as follows:

$$\mathrm{D}_G(x', y'; x, y) = \mathrm{D}_F(x'; x) + \frac{r}{2}\|\mathbf{A}(x' - x) + \gamma\mathbf{W}'(y' - y)\|^2$$

$$\overset{(a)}{\leq} \frac{L_f}{2}\|x' - x\|^2 + \frac{r}{2}\|\mathbf{A}(x' - x) + \gamma\mathbf{W}'(y' - y)\|^2$$

$$\overset{(b)}{\leq} \frac{L_f}{2}\|x' - x\|^2 + r\|\gamma\mathbf{W}'(y' - y)\|^2 + r\|\mathbf{A}(x' - x)\|^2$$

$$\overset{(c)}{\leq} \frac{L_f}{2}\|x' - x\|^2 + r\gamma^2 L_{\mathbf{W}'}\|y' - y\|^2 + rL_{\mathbf{A}}\|x' - x\|^2$$

$$\overset{(d)}{=} \frac{L_f + \mu_f}{2}\|x' - x\|^2 + \frac{\mu_f\gamma^2 L_{\mathbf{W}'}}{2L_{\mathbf{A}}}\|y' - y\|^2,$$

$$\overset{(e)}{\leq} \frac{1}{2}\max\left\{L_f + \mu_f, \mu_f\frac{\mu_{\mathbf{A}} + L_{\mathbf{A}}}{L_{\mathbf{A}}}\frac{L_{\mathbf{W}'}}{\mu_{\mathbf{W}'}}\right\}\left\|\begin{pmatrix}x' - x \\ y' - y\end{pmatrix}\right\|^2,$$

where (a) is due to Assumption 1; (b) is due to Young's inequality; (c) is due to Assumption 2 and eq. (5); (d) and (e) is due to eq. (12). $\qquad\square$

### B.2 PROOF OF LEMMA 2

*Proof.* To obtain the formula for $L_{\mathbf{B}}$, consider an arbitrary $z \in (\mathbb{R}^m)^n$:

$$\|\mathbf{B}^\top z\|^2 = \|\mathbf{A}^\top z\|^2 + \|\gamma\mathbf{W}'z\|^2$$

$$\overset{(a)}{\leq} (L_{\mathbf{A}} + \gamma^2 L_{\mathbf{W}'})\|z\|^2$$

$$\overset{(b)}{=} \left(L_{\mathbf{A}} + (L_{\mathbf{A}} + \mu_{\mathbf{A}})\frac{L_{\mathbf{W}'}}{\mu_{\mathbf{W}'}}\right)\|z\|^2,$$

where (a) is due to Assumption 2 and eq. (5); (b) is due to eq. (12).

To derive the formula for $\mu_{\mathbf{B}}$, first of all, note that by eq. (8)

$$(\ker\mathbf{B}^\top)^\perp = \mathrm{range}\,\mathbf{B} = \mathrm{range}\,\mathbf{A} + \mathrm{range}\,\mathbf{W}' = \mathrm{range}\,\mathbf{A} + \mathcal{L}_m^\perp. \qquad (26)$$

Let $z \in (\ker\mathbf{B}^\top)^\perp = u + v$, where $u = (u_1, \ldots, u_n)$, $v = (v_0, \ldots, v_0) \in (\mathbb{R}^m)^n$ such that $u \in \mathcal{L}_m^\perp$ and $v \in \mathcal{L}_m$.

We can show that $v_0 \in \text{range}\, \mathbf{S}$. In order to do that, let us show that $\langle v_0, w_0 \rangle = 0$ for all $w_0 \in \ker \mathbf{S}$. Let $w = (w_0, \ldots, w_0) \in \mathcal{L}_m$. The fact that $w_0 \in \ker \mathbf{S}$ and $w \in \mathcal{L}_m$ implies $w \in \ker \mathbf{A}\mathbf{A}^\top = \ker \mathbf{A}^\top$. Hence, it is easy to show that $w \in \ker \mathbf{B}^\top = (\text{range}\, \mathbf{B})^\perp$. Then, we obtain

$$n\langle v_0, w_0 \rangle \stackrel{(a)}{=} \langle v, w \rangle \stackrel{(b)}{=} \langle u + v, w \rangle = \langle z, w \rangle \stackrel{(c)}{=} 0,$$

where (a) follows from the definition of $v$ and $w$; (b) follows from the fact that $u \in \mathcal{L}_m^\perp$ and $w \in \mathcal{L}_m$; (c) follows from the fact that $z \in \text{range}\, \mathbf{B}$ and $w \in (\text{range}\, \mathbf{B})^\perp$. Hence, $v_0 \in \text{range}\, \mathbf{S}$.

Further, we get

$$
\begin{aligned}
\|\mathbf{B}^\top z\|^2 &\stackrel{(a)}{=} \|\mathbf{A}^\top(u+v)\|^2 + \|\gamma \mathbf{W}'(u+v)\|^2 \\
&\stackrel{(b)}{=} \|\mathbf{A}^\top(u+v)\|^2 + \|\gamma \mathbf{W}'u\|^2 \\
&\stackrel{(c)}{\geq} \|\mathbf{A}^\top(u+v)\|^2 + \gamma^2 \mu_{\mathbf{W}'}\|u\|^2 \\
&= \|\mathbf{A}^\top u\|^2 + \|\mathbf{A}^\top v\|^2 + 2\langle \mathbf{A}^\top u, \mathbf{A}^\top v \rangle + \gamma^2 \mu_{\mathbf{W}'}\|u\|^2 \\
&\stackrel{(d)}{\geq} -\|\mathbf{A}^\top u\|^2 + \frac{1}{2}\|\mathbf{A}^\top v\|^2 + \gamma^2 \mu_{\mathbf{W}'}\|u\|^2 \\
&\stackrel{(e)}{=} -\|\mathbf{A}^\top u\|^2 + \frac{1}{2}\langle v_0, n\mathbf{S}v_0 \rangle + \gamma^2 \mu_{\mathbf{W}'}\|u\|^2 \\
&\stackrel{(f)}{\geq} -L_\mathbf{A}\|u\|^2 + \frac{n\mu_\mathbf{A}}{2}\|v_0\|^2 + \gamma^2 \mu_{\mathbf{W}'}\|u\|^2 \\
&= -L_\mathbf{A}\|u\|^2 + \frac{\mu_\mathbf{A}}{2}\|v\|^2 + \gamma^2 \mu_{\mathbf{W}'}\|u\|^2 \\
&\stackrel{(g)}{=} \frac{\mu_\mathbf{A}}{2}\|v\|^2 + \mu_\mathbf{A}\|u\|^2 \\
&\stackrel{(h)}{\geq} \frac{\mu_\mathbf{A}}{2}\|z\|^2,
\end{aligned}
$$

where (a) and (h) is due to the definitions of $u$ and $v$; (b) is due to the fact that $v \in \mathcal{L}_m$; (c) is due to eq. (5) and eq. (8); (d) uses Young's inequality; (e) is due to the definitions of $v$ and $\mathbf{S}$, and

$$\|\mathbf{A}^\top v\|^2 = \left\| \begin{pmatrix} \mathbf{A}_1^\top v_0 \\ \vdots \\ \mathbf{A}_n^\top v_0 \end{pmatrix} \right\|^2 = \sum_{i=1}^n \|\mathbf{A}_i^\top v_0\|^2 = \langle v_0, \sum_{i=1}^n \mathbf{A}_i \mathbf{A}_i^\top v_0 \rangle = \langle v_0, n\mathbf{S}v_0 \rangle;$$ (f) is due to Assumption 2 and the definition of $v$; (g) is due to eq. (12).

$\square$

### B.3 PROOF OF THEOREM 1

**Lemma 5** (Salim et al. (2022a), Section 6.3.2). *Let $\mathbf{M}$ be a matrix with $\mu_\mathbf{M} > 0$, $\mathbf{r} \in \text{range}\, \mathbf{M}$ and $\mathbf{M}v_0 = \mathbf{r}$. Then $\mathcal{P}_\mathbf{M}(\mathbf{M}^\top \mathbf{M})(v - v_0) = v - \mathbf{Chebyshev}(v, \mathbf{M}, \mathbf{r})$, where $\mathbf{Chebyshev}$ is defined as Algorithm 6.*

---

**Algorithm 6 Chebyshev**$(v, \mathbf{M}, \mathbf{r})$: Chebyshev iteration (Gutknecht and Röllin (2002), Algorithm 4)

---

1: **Parameters:** $v, \mathbf{M}, \mathbf{r}$.
2: $n := \left\lceil \sqrt{\frac{L_\mathbf{M}}{\mu_\mathbf{M}}} \right\rceil$
3: $\rho := \left(L_\mathbf{M} - \mu_\mathbf{M}\right)^2/16, \nu := (L_\mathbf{M} + \mu_\mathbf{M})/2$
4: $\delta^0 := -\nu/2$
5: $p^0 := -\mathbf{M}^\top(\mathbf{M}v - \mathbf{r})/\nu$
6: $v^1 := v + p^0$
7: **for** $i = 1, \ldots, n-1$ **do**
8: $\qquad \beta^{i-1} := \rho/\delta^{i-1}$
9: $\qquad \delta^i := -(\nu + \beta^{i-1})$
10: $\qquad p^i := \left(\mathbf{M}^\top(\mathbf{M}v^i - \mathbf{r}) + \beta^{i-1}p^{i-1}\right)/\delta^i$
11: $\qquad v^{i+1} := v^i + p^i$
12: **end for**
13: **Output:** $v^n$

---

**Proof of Theorem 1**

*Proof.* Applying Lemma 3 to $\mathbf{W}$ and $\mathbf{B}^\top\mathbf{B}$, we derive that, due to eq. (18), it holds

$$\lambda_{\max}^2(\mathbf{W}') \le L_{\mathbf{W}'} = (19/15)^2, \quad \lambda_{\min+}^2(\mathbf{W}') \ge \mu_{\mathbf{W}'} = (11/15)^2, \tag{27}$$

and by eq. (6) the polynomial $\mathcal{P}_\mathbf{W}$ has a degree of $\lceil\sqrt{\kappa_\mathbf{W}}\rceil$. Similarly, due to eq. (19), it holds

$$\sigma_{\max}^2(\mathbf{K}) = \lambda_{\max}(\mathbf{K}^\top\mathbf{K}) \le L_\mathbf{K} = 19/15, \quad \sigma_{\min+}^2(\mathbf{K}) = \lambda_{\min+}(\mathbf{K}^\top\mathbf{K}) \ge \mu_\mathbf{K} = 11/15, \tag{28}$$

and since $\kappa_\mathbf{B} = \frac{L_\mathbf{B}}{\mu_\mathbf{B}}$, the polynomial $\mathcal{P}_\mathbf{B}$ has a degree of $\lceil\sqrt{\kappa_\mathbf{B}}\rceil$.

We implement computation of the term $\mathbf{K}^\top(\mathbf{K}u - \mathbf{b}')$ in line 6 of Algorithm 1 via Algorithm 5 by Lemma 5:

$$\mathbf{K}^\top(\mathbf{K}u - \mathbf{b}') = \mathbf{K}^\top\mathbf{K}(u - u_0) = \mathcal{P}_\mathbf{B}(\mathbf{B}^\top\mathbf{B})(u - u_0)$$
$$= u - \mathbf{Chebyshev}(u, \mathbf{B}, \mathbf{b}) = \mathbf{K\_Chebyshev}(u).$$

Similarly, utilizing Lemma 5, we get

$$\mathbf{W}'y = \mathcal{P}_{\sqrt{\mathbf{W}}}(\mathbf{W})y = \mathcal{P}_{\sqrt{\mathbf{W}}}(\sqrt{\mathbf{W}}^\top\sqrt{\mathbf{W}})(y - 0) = y - \mathbf{Chebyshev}(y, \sqrt{\mathbf{W}}, 0) = \mathbf{mulW}'(y), \tag{29}$$

where $\mathbf{mulW}'$ is defined as Algorithm 3.

Therefore, Algorithm 2 is equivalent to Algorithm 1.

From eqs. (13) and (27), $\frac{\mu_\mathbf{A}+L_\mathbf{A}}{L_\mathbf{A}} \le 2$ and $(19/11)^2 \le 3$, we get

$$L_G = \max\left\{L_f + \mu_f, \mu_f\frac{\mu_\mathbf{A} + L_\mathbf{A}}{L_\mathbf{A}}\frac{L_{\mathbf{W}'}}{\mu_{\mathbf{W}'}}\right\} \le \mu_f\max\{1 + \kappa_f, 6\}, \tag{30}$$

$$\mu_G = \mu_f\min\left\{\frac{1}{2}, \frac{\mu_\mathbf{A} + L_\mathbf{A}}{4L_\mathbf{A}}\right\} \ge \frac{\mu_f}{4}, \tag{31}$$

$$\kappa_G = \frac{L_G}{\mu_G} \le 4\max\{1 + \kappa_f, 6\}. \tag{32}$$

From eqs. (15) and (27) we get

$$\kappa_\mathbf{B} = \frac{L_\mathbf{B}}{\mu_\mathbf{B}} \le 2\left(\kappa_\mathbf{A} + (19/11)^2(1 + \kappa_\mathbf{A})\right) \le 8\kappa_\mathbf{A} + 6. \tag{33}$$

From eq. (28) we obtain

$$\kappa_\mathbf{K} = \frac{L_\mathbf{K}}{\mu_\mathbf{K}} = 19/11, \tag{34}$$

and substituting eqs. (32) and (33) to Proposition 1, we obtain as its direct corollary that $k = O(\sqrt{\kappa_f} \log(1/\varepsilon))$. Each iteration of Algorithm 2 require $O(1)$ computations of $\nabla F$, $O(\sqrt{\kappa_\mathbf{B}}) = O(\sqrt{\kappa_\mathbf{A}})$ multiplications by $\mathbf{A}, \mathbf{A}^\top$ and $O(\sqrt{\kappa_\mathbf{A}}\sqrt{\kappa_\mathbf{W}})$ multiplications by $\mathbf{W}$, which gives us the statement of Theorem 1. The values of the parameters $\tau, \eta, \theta, \alpha$ in Theorem 1 are derived from Proposition 1 as follows. We have $\tau = \min\left\{1, \frac{1}{2}\sqrt{\frac{\kappa_\mathbf{K}}{\kappa_G}}\right\} = \min\left\{1, \frac{1}{2}\sqrt{\frac{19}{44 \max\{1+\kappa_f, 6\}}}\right\}$ due to eqs. (32) and (34); $\eta = \frac{1}{4\tau L_G} = \frac{1}{4\tau \max\{L_f + \mu_f, 6\mu_f\}}$ due to eq. (30); $\theta = \frac{15}{19\eta}$ due to eq. (28) and $\alpha = \mu_G = \frac{\mu_f}{4}$ due to eq. (31). $\qquad\square$

## C   PROOF OF THEOREM 2

### C.1   DUAL PROBLEM

Let us construct the lower bound for the problem dual to the initial one. Consider primal problem with zero r.h.s. in constraints

$$\min_{x_1, \ldots, x_n \in \ell_2} \sum_{i=1}^{n} f_i(x_i)$$
$$\text{s.t.} \sum_{i=1}^{n} \mathbf{A}_i x_i = 0. \tag{35}$$

The dual problem has the form

$$\min_{x_1, \ldots, x_n \in \ell_2} \max_z \left[\sum_{i=1}^{n} f_i(x_i) - \langle z, \mathbf{A}_i x_i \rangle\right] = \max_z \left[-\max_{x_1, \ldots, x_n \in \ell_2} \sum_{i=1}^{n} \langle \mathbf{A}_i^\top z, x_i \rangle - f_i(x_i)\right]$$
$$= -\min_z \sum_{i=1}^{n} f_i^*(\mathbf{A}_i^\top z). \tag{36}$$

Introducing local copies of $w$ at each node, we get

$$\min_{z_1, \ldots, z_n} \sum_{i=1}^{n} g_i(z_i) := \sum_{i=1}^{n} f_i^*(\mathbf{A}_i^\top z_i)$$
$$\text{s.t.} \mathbf{W}z = 0. \tag{37}$$

### C.2   EXAMPLE GRAPH

We follow the principle of lower bounds construction introduced in Kovalev et al. (2021) and take the example graph from Scaman et al. (2017). Let the functions held by the nodes be organized into a path graph with $n$ vertices, where $n$ is divisible by 3. The nodes of graph $\mathcal{G} = (\mathcal{V}, \mathcal{E})$ are divided into three groups $\mathcal{V}_1 = \{1, \ldots, n/3\}, \mathcal{V}_2 = \{n/3+1, \ldots, 2n/3\}, \mathcal{V}_3 = \{2n/3+1, \ldots, n\}$ of $n/3$ vertices each.

Now we recall the construction from Scaman et al. (2017). Maximum and minimum eigenvalues of a path graph have form $\lambda_{\max}(W) = 2\left(1 + \cos\frac{\pi}{n}\right)$, $\lambda_{\min+}(W) = 2\left(1 - \cos\frac{\pi}{n}\right)$. Let $\beta_n = \frac{1+\cos\left(\frac{\pi}{n}\right)}{1-\cos\left(\frac{\pi}{n}\right)}$. Since $\beta_n \overset{n\to\infty}{\to} +\infty$, there exists $n = 3m \geq 3$ such that $\beta_n \leq \kappa_\mathbf{W} < \beta_{n+3}$. For this $n$, introduce edge weights $w_{i,i+1} = 1 - a\mathbb{I}\{i=1\}$, take the corresponding weighed Laplacian $W_a$ and denote its condition number $\kappa(W_a)$. If $a = 1$, the network is disconnected and therefore $\kappa(W_a) = \infty$. If $a = 0$, we have $\kappa(W_a) = \beta_n$. By continuity of Laplacian spectra we obtain that for some $a \in [0, 1)$ it holds $\kappa(W_a) = \kappa_\mathbf{W}$. Note that $\pi/(n+3) \in [0, \pi/3]$ for $x \in [0, \pi/3]$ it holds $1 - \cos x \geq x^2/4$. We have

$$\kappa_\mathbf{W} \leq \beta_{n+3} = \frac{1 + \cos\frac{\pi}{n+3}}{1 - \cos\frac{\pi}{n+3}} \leq \frac{72(n+3)^2}{\pi^2} \leq \frac{288n^2}{\pi^2} \leq 32n^2 \quad \Rightarrow \quad \sqrt{\kappa_\mathbf{W}} \leq 4\sqrt{2}n = O(n). \tag{38}$$

## C.3 EXAMPLE FUNCTIONS

We let $e_1 = (1\ 0\ \dots\ 0)^\top$ denote the first coordinate vector and define functions

$$f_i(p, t) = \frac{\mu_f}{2} \left\| p + \frac{\sqrt{\hat{L}_\mathbf{A}}}{2\mu_f} e_1 \right\|^2 + \frac{L_f}{2} \|t\|^2. \tag{39}$$

We immediately note that each $f_i$ is $L_f$-smooth and $\mu_f$-strongly convex. The first term has the form $\frac{\mu_f}{2}\|p + ce_1\|^2$, and its convex conjugate is $\left(\frac{\mu_f}{2}\|p + ce_1\|^2\right)^* = \max_p \left\{ \langle u, p \rangle - \frac{\mu_f}{2}\|p + ce_1\|^2 \right\}$. The gradient by $p$ is $u - \mu_f(p + ce_1) = 0$, thus $p = \frac{1}{\mu_f}u - ce_1$ and $\max_p \left\{ \langle u, p \rangle - \frac{\mu_f}{2}\|p + ce_1\|^2 \right\} = \frac{\|u\|^2}{2\mu_f} - cu_1$. Correspondingly,

$$f_i^*(u, v) = \frac{1}{2\mu_f}\|u\|^2 + \frac{1}{2L_f}\|v\|^2 - \frac{\sqrt{\hat{L}_\mathbf{A}}}{2\mu_f}u_1.$$

To define matrices $\mathbf{A}_i$, we first introduce

$$\mathbf{E}_1 = \begin{pmatrix} 1 & 0 & 0 & 0 & 0 & \dots \\ 0 & 1 & -1 & 0 & 0 & \dots \\ 0 & 0 & 0 & 0 & 0 & \dots \\ 0 & 0 & 0 & 1 & -1 & \dots \\ \vdots & \vdots & \vdots & \vdots & \vdots & \ddots \end{pmatrix}, \quad \mathbf{E}_2 = \begin{pmatrix} 1 & -1 & 0 & 0 & 0 & \dots \\ 0 & 0 & 0 & 0 & 0 & \dots \\ 0 & 0 & 1 & -1 & 0 & \dots \\ 0 & 0 & 0 & 0 & 0 & \dots \\ \vdots & \vdots & \vdots & \vdots & \vdots & \ddots \end{pmatrix}.$$

Let $\hat{L}_\mathbf{A} = \frac{1}{2}L_\mathbf{A} - \frac{3}{4}\mu_\mathbf{A}$, $\hat{\mu}_\mathbf{A} = \frac{3}{2}\mu_\mathbf{A}$ and introduce

$$\mathbf{A}_i = \begin{cases} [\sqrt{\hat{L}_\mathbf{A}}\mathbf{E}_1^\top \quad \sqrt{\hat{\mu}_\mathbf{A}}\mathbf{I}], & i \in \mathcal{V}_1 \\ [\ \mathbf{0} \qquad\quad \mathbf{0}\ ], & i \in \mathcal{V}_2 \\ [\sqrt{\hat{L}_\mathbf{A}}\mathbf{E}_2^\top \quad \sqrt{\hat{\mu}_\mathbf{A}}\mathbf{I}], & i \in \mathcal{V}_3 \end{cases}$$

Let us make sure that the choice of $\mathbf{A}_i$ guarantees constants $L_\mathbf{A}, \mu_\mathbf{A}$ from (4).

$$\max_i \lambda_{\max}(\mathbf{A}_i\mathbf{A}_i^\top) = \lambda_{\max}\left(\hat{L}_\mathbf{A}\mathbf{E}_1^\top\mathbf{E}_1 + \hat{\mu}_\mathbf{A}\mathbf{I}\right) = 2\hat{L}_\mathbf{A} + \hat{\mu}_\mathbf{A} = L_\mathbf{A},$$

$$\lambda_{\min}^+\left(\frac{1}{n}\sum_{i=1}^n \mathbf{A}_i\mathbf{A}_i^\top\right) = \lambda_{\min}^+\left(\frac{1}{3}(\hat{L}_\mathbf{A}\mathbf{E}_1^\top\mathbf{E}_1 + \hat{\mu}_\mathbf{A}\mathbf{I}) + \frac{1}{3}(\hat{L}_\mathbf{A}\mathbf{E}_2^\top\mathbf{E}_2 + \hat{\mu}_\mathbf{A}\mathbf{I})\right)$$

$$= \frac{2}{3}\hat{\mu}_\mathbf{A} = \mu_\mathbf{A}.$$

Let $\widetilde{\mathbf{M}} = \begin{pmatrix} 1 & -1 \\ -1 & 1 \end{pmatrix}$ and

$$\mathbf{M}_1 = \mathbf{E}_1^\top\mathbf{E}_1 = \begin{pmatrix} 1 & 0 & 0 & \dots \\ 0 & \widetilde{\mathbf{M}} & 0 & \dots \\ 0 & 0 & \widetilde{\mathbf{M}} & \dots \\ \vdots & \vdots & \vdots & \ddots \end{pmatrix}, \quad \mathbf{M}_2 = \mathbf{E}_2^\top\mathbf{E}_2 = \begin{pmatrix} \widetilde{\mathbf{M}} & 0 & 0 & \dots \\ 0 & \widetilde{\mathbf{M}} & 0 & \dots \\ 0 & 0 & \widetilde{\mathbf{M}} & \dots \\ \vdots & \vdots & \vdots & \ddots \end{pmatrix}$$

The dual functions take the form

$$g_i(z) = f_i^*(\mathbf{A}_i^\top z) = \begin{cases} \frac{1}{2\mu_f}\|\sqrt{\hat{L}_\mathbf{A}}\mathbf{E}_1 z\|^2 + \frac{1}{2L_f}\|\sqrt{\hat{\mu}_\mathbf{A}}z\|^2 - \frac{\hat{L}_\mathbf{A}}{2\mu_f}z_1, & i \in \mathcal{V}_1 \\ 0, & i \in \mathcal{V}_2 \\ \frac{1}{2\mu_f}\|\sqrt{\hat{L}_\mathbf{A}}\mathbf{E}_2 z\|^2 + \frac{1}{2L_f}\|\sqrt{\hat{\mu}_\mathbf{A}}z\|^2 - \frac{\hat{L}_\mathbf{A}}{2\mu_f}z_1, & i \in \mathcal{V}_3 \end{cases}$$

$$= \begin{cases} \frac{1}{2}z^\top\left(\frac{\hat{L}_\mathbf{A}}{\mu_f}\mathbf{M}_1 + \frac{\hat{\mu}_\mathbf{A}}{L_f}\mathbf{I}\right)z - \frac{\hat{L}_\mathbf{A}}{2\mu_f}z_1, & i \in \mathcal{V}_1 \\ 0, & i \in \mathcal{V}_2 \\ \frac{1}{2}z^\top\left(\frac{\hat{L}_\mathbf{A}}{\mu_f}\mathbf{M}_2 + \frac{\hat{\mu}_\mathbf{A}}{L_f}\mathbf{I}\right)z - \frac{\hat{L}_\mathbf{A}}{2\mu_f}z_1, & i \in \mathcal{V}_3 \end{cases} \tag{40}$$

Therefore, we have

$$
\begin{aligned}
\sum_{i=1}^n g_i(z) &= \frac{n}{3}\left[\frac{\hat{L}_{\mathbf{A}}}{2\mu_f}z^\top(\mathbf{M}_1+\mathbf{M}_2)z + \frac{\hat{\mu}_{\mathbf{A}}}{L_f}z^\top z - \frac{\hat{L}_{\mathbf{A}}}{\mu_f}z_1\right] \\
&= \frac{n}{3}\frac{\hat{L}_{\mathbf{A}}}{\mu_f}\left[\frac{1}{2}z^\top\mathbf{M}z - z_1 + \frac{\hat{\mu}_{\mathbf{A}}\mu_f}{\hat{L}_{\mathbf{A}}L_f}z^\top z\right],
\end{aligned}
\tag{41}
$$

where

$$
\mathbf{M} = \mathbf{M}_1 + \mathbf{M}_2 = \begin{pmatrix} 2 & -1 & 0 & 0 & 0 & \dots \\ -1 & 2 & -1 & 0 & 0 & \dots \\ 0 & -1 & 2 & -1 & 0 & \dots \\ \vdots & \vdots & \vdots & \vdots & \vdots & \ddots \end{pmatrix}.
$$

Now we formulate the lower complexity bounds for $\sum_{i=1}^n g_i(z)$, where $g_i(z)$ are defined in (40).

### C.4 DERIVING THE LOWER BOUND

**Lemma 6.** *Function $\sum_{i=1}^n g_i(z)$ attains its minimum at $z^* = \{\rho^k\}_{k=1}^\infty$, where*

$$
\rho = \frac{\sqrt{\frac{2L_{\mathbf{A}}L_f}{\mu_{\mathbf{A}}\mu_f}+1}-1}{\sqrt{\frac{2L_{\mathbf{A}}L_f}{\mu_{\mathbf{A}}\mu_f}+1}+1}.
$$

*Proof.* In the lower bound example in (Lemma 1 in Appendix C) Kovalev et al. (2021) it was shown that function

$$
h(z) = \frac{1}{2}z^\top\mathbf{M}z + \frac{3\mu}{L-\mu}z^\top z - z_1
$$

attains its minimum at $z_k^* = \rho^k$, where

$$
\rho = \frac{\sqrt{\frac{2L}{3\mu}+\frac{1}{3}}-1}{\sqrt{\frac{2L}{3\mu}+\frac{1}{3}}+1}.
$$

Let us deduce the expression for $L/\mu$ in terms of $L_{\mathbf{A}}, L_f, \mu_{\mathbf{A}}, \mu_f$. We enforce $h(z) = \sum_{i=1}^n g_i(z)$ and set

$$
\frac{3\mu}{L-\mu} = \frac{\mu_{\mathbf{A}}\mu_f}{L_{\mathbf{A}}L_f} \Rightarrow \frac{L}{\mu} = 1 + 3\frac{L_{\mathbf{A}}L_f}{\mu_{\mathbf{A}}\mu_f}.
$$

Therefore, $h(z)$ attains its minimum at $z^k = \rho^k$, where

$$
\rho = \frac{\sqrt{\frac{2L_{\mathbf{A}}L_f}{\mu_{\mathbf{A}}\mu_f}+1}-1}{\sqrt{\frac{2L_{\mathbf{A}}L_f}{\mu_{\mathbf{A}}\mu_f}+1}+1}.
$$

$\square$

Let us first show the lower bound on the number of communications. Without loss of generality we can assume that the initial point chosen by a first-order algorithm is $x^0 = 0$ (otherwise we can shift the variables accordingly), thus $\mathcal{M}_i(0) = \{0\}$, $\mathcal{H}_i(0) = \{0\}$.

**Lemma 7.** *Let $s_i(k)$ denote the maximum index of a nonzero component of vector blocks $p, t$ held by $i$-th node at step $k$ in its local memory $\mathcal{M}_i(k)$ and vector $z$ held by $i$-th node in its local memory $\mathcal{H}_i(k)$, i.e.*

$$
s_i(k) = \begin{cases} 0, & \text{if } \mathcal{M}_i(k) \subseteq \{0\} \text{ and } \mathcal{H}_i(k) \subseteq \{0\} \\ \min\Big\{s \in \{1,2,\dots\} : \mathcal{H}_i(k) \subseteq \mathrm{Span}\,\{e_1,\dots,e_s\} \\ \quad \text{and } \mathcal{M}_i(k) \subseteq \mathrm{Span}\,\{e_1,\dots,e_s\} \times \{e_1,\dots,e_s\}\Big\}, & \text{else.} \end{cases}
$$

*Let $k_q$ denote the number of algorithm step by which exactly $q$ communication steps have been performed, where $q \geq 0$. For any $k \in \{1, \ldots, k_q\}$ we have*

$$\max_i s_i(k) \leq 2 + \left\lfloor \frac{q}{\frac{n}{3} + 1} \right\rfloor \tag{42}$$

*Proof.* If the method performs a multiplication by $\mathbf{A}_i$, it transfers the information from $\mathcal{M}_i(k)$ to $\mathcal{H}_i(k)$. If multiplication by $\mathbf{A}_i^\top$ is performed, the information is transferred in the opposite direction, i.e. from $\mathcal{H}_i(k)$ to $\mathcal{M}_i(k)$. If the method performs a matrix multiplication step (either by $\mathbf{A}$ or $\mathbf{A}^\top$), then from the structure of $\mathbf{A}_i$ we obtain

$$s_i(k+1) \leq s_i(k) + \begin{cases} 1 - (s_i(k) \mod 2), & i \in \mathcal{V}_1 \\ 0, & i \in \mathcal{V}_2 \\ (s_i(k) \mod 2), & i \in \mathcal{V}_3 \end{cases}$$

We will prove that
1. For $q = 2\ell(n/3 + 1)$, $\ell \in \{0, 1, \ldots\}$ we have

$$s_i(k_q) \leq \begin{cases} 1 + 2\ell, & i \in \mathcal{V}_1 \\ 1 + 2\ell, & i \in \mathcal{V}_2 \\ 2 + 2\ell, & i \in \mathcal{V}_3 \end{cases} \tag{43}$$

2. For $q = (2\ell + 1)(n/3 + 1)$, $\ell \in \{0, 1, \ldots\}$ we have

$$s_i(k_q) \leq \begin{cases} 2 + (2\ell + 1), & i \in \mathcal{V}_1 \\ 1 + (2\ell + 1), & i \in \mathcal{V}_2 \\ 1 + (2\ell + 1), & i \in \mathcal{V}_3 \end{cases} \tag{44}$$

The proof follows by induction.

**Induction basis**. Let $q = 0$. From definitions of $f_i$ and $\mathbf{A}_i$ it follows that

$$s_i(k_0) \leq \begin{cases} 1, & i \in \mathcal{V}_1 \\ 0, & i \in \mathcal{V}_2 \\ 2, & i \in \mathcal{V}_3 \end{cases}$$

Therefore, for $q = 0$ our statement holds.

**Induction step for** $q = (2\ell + 1)(n/3 + 1)$. Consider $q_- = q - n/3 = 2\ell(n/3 + 1)$. From (43) we have that for the spread of nonzero components from $\mathcal{V}_3$ to $\mathcal{V}_1$ it requires $n/3$ communication rounds to reach node $n/3 + 1$. After one more communication round, the information reaches node $n/3$.

**Induction step for** $q = 2\ell(n/3 + 1)$. The proof follows by the same argument as for $q = (2\ell + 1)(n/3 + 1)$.

We just proved the statement of lemma, i.e. relation (42), for $q$ divisible by $(n/3 + 1)$. Between such checkpoints, the information (i.e. the number of nonzero components) traverses nodes of $\mathcal{V}_2$ and therefore $\max_i s_i(k)$ stays unchanged. Thus the statement of the lemma is proven. $\qquad\square$

Now we estimate the distance to optimum. Due to the strong duality, the solution of problem (35) can be obtained from the solution of its dual (36) as

$$x_i^*(z^*) = \binom{p_i}{t_i}(\mathbf{A}_i^\top z^*) = \begin{pmatrix} \frac{\sqrt{\hat{L}_\mathbf{A}}}{\mu_f}(E_{(i)}z^* - e_1) \\ \frac{\sqrt{\hat{\mu}_\mathbf{A}}}{L_f}z^* \end{pmatrix}.$$

Therefore, denoting $\alpha = \frac{\hat{\mu}_\mathbf{A}}{L_f^2}$,

$$\|x_i^k - x_i^*\|^2 \geq \|t_i^k - t_i^*\|^2 \geq \sum_{\ell = s_i(k)+1}^\infty (t_{i,\ell}^*)^2 = \alpha \sum_{\ell = s_i(k)+1}^\infty (z_\ell^*)^2 = \alpha \sum_{\ell = s_i(k)+1}^\infty \rho^{2\ell}$$

$$= \alpha \frac{\rho^{2s_i(k)+2}}{1 - \rho^2} = \alpha \frac{\rho^{6 + 2\lfloor \frac{q}{n/3+1} \rfloor}}{1 - \rho^2} \overset{(a)}{\geq} \alpha \frac{\rho^{6 + \frac{2q}{2n/3}}}{1 - \rho^2} = \alpha \frac{\rho^6}{1 - \rho^2} \cdot \rho^{\frac{3q}{n}} \overset{(b)}{\geq} \alpha \frac{\rho^6}{1 - \rho^2} \cdot \rho^{\frac{12q\sqrt{2}}{\sqrt{\kappa}\mathbf{W}}},$$

where (a) holds since $n/3 \geq 1$; (b) holds due to (38).

Following Kovalev et al. (2021), we obtain that

$$\rho \geq \max\left(0, 1 - 3\sqrt{\frac{2\mu_{\mathbf{A}}\mu_f}{L_{\mathbf{A}}L_f}}\right).$$

Therefore,

$$\|x_i^k - x_i^*\|_2^2 \geq \alpha \frac{\rho^6}{1-\rho^2}\left(\max\left(0, 1 - 3\sqrt{\frac{2\mu_{\mathbf{A}}\mu_f}{L_{\mathbf{A}}L_f}}\right)\right)^{\frac{3q}{\sqrt{\kappa_{\mathbf{W}}}}}.$$

It follows that the number of communications is lower bounded as

$$N_{\mathbf{W}} \geq \Omega\left(\sqrt{\kappa_{\mathbf{W}}}\sqrt{\frac{L_{\mathbf{A}}L_f}{\mu_{\mathbf{A}}\mu_f}}\log\left(\frac{1}{\varepsilon}\right)\right), \tag{45}$$

and the number of matrix multiplications at each node is lower bounded as

$$N_{\mathbf{A}} \geq \Omega\left(\sqrt{\frac{L_{\mathbf{A}}L_f}{\mu_{\mathbf{A}}\mu_f}}\log\left(\frac{1}{\varepsilon}\right)\right). \tag{46}$$

### C.5 LOWER BOUND ON THE NUMBER OF GRADIENT COMPUTATIONS

To get the lower bound on local gradient calls, let us consider a problem

$$\min_{\substack{x_1,\ldots x_n\in\mathbb{R}^d \\ u_1,\ldots,u_n\in\mathbb{R}^d}} \quad \sum_{i=1}^n f_i(x_i) + \sum_{i=1}^n v_i(u_i)$$

$$\text{s.t.} \quad \sum_{i=1}^n \mathbf{A}_i x_i = 0 \tag{47}$$

where all $f_i(x)$ are defined in (39) and all $v_i(u_i)$ are similar and defined as

$$v_i(u_i) = \frac{L_f}{8}u_i^\top \mathbf{M} u_i + \frac{\mu_f}{2}u^\top u - \frac{L_f}{4}u_1.$$

First, note that each $v_i(u_i)$ is $L_f$-smooth and $\mu_f$-strongly convex, i.e. problem (47) satisfies the assumptions of Theorem 2.

Problem (47) falls into two independent parts. The first part is minimization of $\sum_{i=1}^n f_i(x_i)$ subject to constraints and is identical to (35), while the second part is minimization of $\sum_{i=1}^n v_i(u_i)$ without constraints. Therefore, the lower bounds on the number of communications and number of matrix multiplications for problem (47) are inherited from lower bounds for problem (35) and are the given by $N_{\mathbf{W}}$ in (45) and $N_{\mathbf{A}}$ in (46). It remains to lower bound the number of oracle calls for minimization of $\sum_{i=1}^n v_i(u_i)$.

The structure of $v_i(u_i)$ is the same as the structure of $\sum_{j=1}^n g_j(z)$ defined in (41). Therefore, the minimum of each $v_i(u_i)$ is attained at the same point $u_i^* = u^*$, and the $k$-th component of $u^*$ is given by $(u^*)_k = \nu^k$, where

$$\nu = \frac{\sqrt{\frac{L_f}{\mu_f} + \frac{2}{3}} - 1}{\sqrt{\frac{L_f}{\mu_f} + \frac{2}{3}} + 1}.$$

Since there is no communication constraint on $u_i$, each node runs optimization process individually. Following the same arguments as for function $\sum_{j=1}^n g_j(z)$, we get the lower bound on the number of oracle calls

$$N_f \geq \Omega\left(\sqrt{\frac{L_f}{\mu_f}}\log\left(\frac{1}{\varepsilon}\right)\right).$$

