# OpenReview forum: "Decentralized Optimization with Coupled Constraints"
_ICLR.cc/2025/Conference — ICLR 2025 Poster_

### Official Review · Reviewer_qaH7 · 2024-10-19

**Soundness:** 3
**Presentation:** 2
**Contribution:** 3
**Rating:** 5
**Confidence:** 3

**Summary:**

In this paper, the authors propose an algorithm for decentralized optimization with coupled constrains. Compared by most methods with proximal oracle, it is a first-order approach with a lower computational burden. The algoorithm is motivated on Chebyshev acceleration and Proximal Alternating Predictor-Corrector. The authors also provide the convergence analysis of the proposed algorithm in strongly-convex scenario and also proves that the proposed algorithm can reach the lower bound.

**Strengths:**

1. first order iterations.
2. match the lower bound.
3. suitable for different kinds of problems.

**Weaknesses:**

1. The presentation need to be improved. The contribution of the paper should be highlighted. And maybe it is more friendly for readers if the main algorithm is in Section 4, followed by the introduction of Chebyshev acceleration and Proximal Alternating Predictor-Corrector.

2. It seems that some existing results like [1] do not need the strongly-convex assumption. And the author fails to provide the convergence analysis without strongly-convex assumption.

3.  More experiments should be involved, including more problems with different coupled constraints. Some necessary comparison with algorithms developed for one kinds of constraints, like EXTRA in consensus optimization, should also be taken.

[1] Distributed Optimization With Coupling Constraints.

[2] EXTRA: An Exact First-Order Algorithm for Decentralized Consensus Optimization

**Questions:**

See weakness.

---

> ### Author Response · Authors · 2024-12-04
>
> Dear Reviewer,
>
> We are grateful for your review.
>
> **Highlight the contribution** and **Reduction to decentralized optimization**.
>
> We compared our method to EXTRA and found that it is competitive and even outperforms EXTRA (see plots at https://ibb.co/gj1Tdck). Please also see the common answer to all reviewers.
>
> **Only strongly convex case considered**.
>
> In the case of non-strongly convex objectives it is much easier to recover convergence rates which correspond to the classical unconstrained optimization. For example, the $O(1/k)$ rate from [1] can be achieved by considering the simplified version of problem (10) without the augmented Lagrangian term: $F(x) \to \min_{x, y} \quad \text{s.t.} \quad Ax + Wy = b$, reformulating it as the saddle-point problem $\max_z\min_{x,y} F(x) + \langle z, Ax+ Wy - b \rangle$ and applying the Extragradient method (Korpelevich, 1976) to it. There are many results with sublinear convergence rates for this type of problem, therefore we were not interested in the non-strongly convex setup.
>
> In contrast, it is not that easy to achieve linear convergence (which is natural for gradient descent in strongly convex case) for coupled constraints as indicated by lack of such results in the literature. We achieved it by using the augmented Lagrangian trick and deriving the upper bound on the condition number of the matrix $B^TB$ in Lemma 2, which must be expressed in terms of $\kappa_W$ and $\kappa_A$ from the initial assumptions. The approach might seem simple, but it was not on the surface. Also the derivation of the upper bound turned out to be tricky, although the final proof looks pretty compact.

---

### Official Review · Reviewer_ieaZ · 2024-10-23

**Soundness:** 3
**Presentation:** 3
**Contribution:** 2
**Rating:** 6
**Confidence:** 4

**Summary:**

This paper studies decentralized optimization with (affine) coupled constraints. This general formulation encompasses different applications, from energy networks to vertical (model-partitioned) federated learning. A complexity lower bound is given for this problem, as well as a matching (and thus optimal) algorithm. Both the algorithm and the lower bound are obtained through a reduction to the standard decentralized case, which is usually solved by introducing coupled constraints (the parameters from different nodes should be equal), and solving the constrained problem with a (primal-)dual approach. As such, the algorithm is based on APAPC (Salim et. al., 2022), and the lower bound directly proceeds from standard lower bounds. Adding extra coupling constraints is extremely natural in this case, and it is not surprising that using the same approach as for decentralized unconstrained optimization leads to similar results. The only care should be taken in how these extra constraints are added to ensure that the right quantities are communicated. Experiments demonstrate the (expected) superiority of the approach.

**Strengths:**

- Tight analysis (optimal algorithm + lower bound)
- Efficient off-the-shelf algorithm
- Authors clearly state their contributions and how they leverage previous work

I believe the main strength of the paper is that it uses the right tools to close a gap in the existing literature, so that non-experts can then use the algorithms.

**Weaknesses:**

- Rather incremental contribution. All results directly derive from existing ones on the reformulation from (10), which is rather natural (though not straightforward).

In the end, I have mixed feelings about this paper. The contribution is not particularly innovative or technically challenging, but it has the merit of existing, and gives a well-rounded solution to this problem that people can then use. Therefore, I believe it should be published somewhere, though I am not completely sure this is the right venue, which is why I slightly lean towards acceptance. However, I am not too familiar with ICLR standards so I am looking forward to the discussion phase.

**Questions:**

- Could you precise the results obtained on vertical federated learning (giving a simple corollary in some standard specific case for instance) so that people can compare their results directly with yours? Does it just correspond to $\kappa_A = 1$? How does this result compare with existing work in model-partitioned distributed optimization then?
- Have you tried tuning the value of r in the experiments? What would be the impact?

---

> ### Author Response · Authors · 2024-12-04
>
> Dear Reviewer,
>
> Thank you for your review and comments.
>
> **Incremental contribution**.
>
> Please see the common answer to Reviewers.
>
> **Vertical federated learning (VFL) and model-partitioned distributed optimization**.
>
> In case of VFL linear regression, as described in lines 457-466, we have very simple objective function with $\mu_f = 2\lambda$ and $L_f = 2\lambda + 1$, and all the information from the dataset is moved out to the coupled constraints. Since $\mathbf A_i = \mathbf F_i~\forall i = 2,\ldots,n$ and $\mathbf F_i$ are feature submatrices that can be arbitrary, we can not have any a-priori bound on $\kappa_{\mathbf A}$, because it is the parameter that characterizes all the complexity related to the dataset.
> 	As for the comparison with other algorithms for decentralized model-partitioned optimization, to the best of our knowledge, there are not much publications to compare with. This is expected due to the lack of gradient-based decentralized algorithms for coupled constraints.
> 	A recent survey on federated learning (FL) [1] includes a review of decentralized algorithms for FL in Section 3.3.3, but none of them deals with vertical FL. There is also a couple of more recent papers on decentralized vertical FL without theoretical guarantees [2], [3], and one paper with a sublinear rate of convergence [4].
>
> [1] Ye, Mang, et al. "Heterogeneous federated learning: State-of-the-art and research challenges." ACM Computing Surveys 56.3 (2023): 1-44.
>
> [2] Celdrán, Alberto Huertas, et al. "De-VertiFL: A Solution for Decentralized Vertical Federated Learning." arXiv preprint arXiv:2410.06127 (2024).
>
> [3] Sánchez Sánchez, Pedro Miguel, et al. "Analyzing the robustness of decentralized horizontal and vertical federated learning architectures in a non-IID scenario." Applied Intelligence (2024): 1-17.
>
> [4] Valdeira, Pedro, et al. "A Multi-Token Coordinate Descent Method for Semi-Decentralized Vertical Federated Learning." arXiv preprint arXiv:2309.09977 (2023).
>
>
> **Tuning of $r$ in experiments**.
>
> Regularization parameter $r$ is chosen according to theory, therefore, it can be viewed as a part of our algorithm. We did not tune $r$ in the simulations, since it could break the convergence.

---

### Official Review · Reviewer_HQPK · 2024-10-30

**Soundness:** 4
**Presentation:** 3
**Contribution:** 3
**Rating:** 8
**Confidence:** 3

**Summary:**

This paper studies the task of decentralized optimization under the coupled constraints setting. This setting is quite general, which recovers a wide range of existing problems such as the consensus problem, optimal exchange problem, etc. Using a multi-communication subroutine, the paper proposed a dual-loop algorithm that provably converges with fast and tight convergence guarantees.

**Strengths:**

The authors provided a solid discussion on the related literature, which clearly explained the relative position of this paper with respect to multiple approaches in the prior works. The introduction also was able to convince me the generalizability and significance of this work.

The theoretical results offered by the authors, if correct, is tight and optimal.

The connection between FL and decentralized optimization has long been established. The coupled constraints setting considered in this paper can be a good starting point for technical understanding of VFL.

**Weaknesses:**

Although the upper bounds and lower bounds are provided in this paper and is shown to be tight. The lower bounds is only applicable to a narrow class of algorithms represented by the proposed algorithm. There are other class of algorithms which could have been discussed.

The proposed algorithm uses a multi-communication algorithm in a dual-loop approach, where a subproblem must be solved with communication. Although the process is accelerated such that it is optimal, there exist studies in the decentralized consensus literature that utilize tools such as gradient tracking, ADMM, etc., and prove convergence with single-loop algorithms. This is also preferable in practice since the communication bottleneck is more apparent for modern optimization tasks.

A more complex experiment with real data and even neural networks would be appreciated for VFL. Though it is understandably omitted since it would not satisfy any of the technical assumptions, it would still be nicer to see.

**Questions:**

Currently, the lower bound is based on the algorithmic structure of the proposed algorithm, where the communication round lower bound is dependent on all three condition numbers. Is there a way to get an algorithm that only requires operations that only depend on their own respective condition number?

---

> ### Author Response · Authors · 2024-12-04
>
> Dear Reviewer,
>
> Thank you for positive evaluation of our work!
>
> **Lower bounds**. The lower bound says that it is not possible to separate condition numbers, at least within the class of vanilla first-order algorithms we considered. This class is a direct extension of the standard classes of algorithms considered for classical optimization by Nesterov [1] and for consensus decentralized optimization by Scaman [2]. It was actually a surprise for us that the communication complexity must depend on $\sqrt{\kappa_{\mathbf A}}$, since initially we were trying to obtain the algorithm with $O(\sqrt{\kappa_f}\sqrt{\kappa_{\mathbf W}}\log(1/\varepsilon))$ bound on the number of communication rounds.  For instance, we knew that such separation of $\sqrt{\kappa_{\mathbf A}}$ is indeed achievable in the simpler case of consensus optimization with additional local affine constraints $A x = b$, which are the same for each node. Note that $O(\sqrt{\kappa_f}\sqrt{\kappa_{\mathbf W}}\log(1/\varepsilon))$ communication complexity is the lower bound of (Scaman, 2017) for consensus optimization, and we already knew that we could not eliminate $\sqrt{\kappa_f}$ in our more general setup.
> In our opinion, the derived bounds are interesting because they highlight the additional complexity brought by coupled constraints.  We also do not believe that extending the class of algorithms to allow, for example, the use of the proximal operator of $f_i(x_i)$ will alter the complexity bounds.
>
> **Multi-consensus**.
>
> In Algorithm 2 of our paper multi-communication along with Chebyshev acceleration can be removed by replacing procedure **mulW'** (Algorithm 3) with a multiplication by $\mathbf{W}$ and replacing procedure **K_Chebyshev** (Algorithm 5) with multiplication ${\mathbf A}^\top({\mathbf A} u - {\mathbf b'})$. As a result, only two multiplications by $\mathbf W$ will be performed at each iteration of the method. At the same time, gradient, communication and matrix multiplication complexities of the method will not be separated. In other words, we can come away from multi-consensus at the cost of increased working time (at least, in theory). To the best of our knowledge, multi-consensus is inevitable to reach optimal complexity bounds in gradient computations and communications simultaneously. This is logical, since with single-step consensus each gradient call corresponds to exactly one communication round.
>
> **Numerical experiments.**
>
> Thank you for suggesting an experiment with VFL. However, we think that working with more complicated experiments, i.e. neural networks, requires a different study. One of the issues is how to compare different algorithms. Since problem parameters are unknown, it is a separate task to align parameters of different methods. Meanwhile, such alignment is needed if we want to run an experiment supporting at least some theory. Summing up, we believe that it is better to focus on convex optimization in experiments. You can find additional comments in the common answer to Reviewers.
>
>
> **References**
>
> [1] Nesterov, Yurii. Introductory lectures on convex optimization: A basic course. Vol. 87. Springer Science & Business Media, 2013.
>
> [2] Scaman, Kevin, et al. "Optimal algorithms for smooth and strongly convex distributed optimization in networks." International conference on machine learning. PMLR, 2017.

---

### Official Review · Reviewer_Rp6n · 2024-10-31

**Soundness:** 3
**Presentation:** 3
**Contribution:** 3
**Rating:** 6
**Confidence:** 4

**Summary:**

This paper studies decentralized optimization with constraints. It establishes the lower bound for this problem, and develops algorithms to attain such lower bound. The main idea is to transform the original constrained decentralized problem into (20), and then rely on the algorithm APAPC to develop the optimal algorithm. Numerical algorithms are established to validate the theoretical findings.

**Strengths:**

This paper is well-written and presents several key strengths:

1. The reformulation from problem (1) to problem (20) is novel and insightful, providing a strong foundation for the development of an optimal algorithm.

2. It establishes the first lower bound for decentralized optimization under affine constraints.

3. The paper also develops an algorithm that attains this lower bound, demonstrating both the tightness of the bound and the optimal complexity of the proposed algorithm.

**Weaknesses:**

However, this paper has several notable weaknesses:

1. The novelty is limited. The core idea for constructing the optimal algorithm draws heavily from the APAPC algorithm (Salim et al. 2022a) and Chebyshev acceleration, a common technique in accelerated algorithms for unconstrained decentralized optimization. The authors combine these two techniques to create the proposed algorithm.

2. Another concern is the practical applicability of the proposed algorithm. While it achieves theoretical optimality, it is significantly more complex to implement than existing baselines. As shown in Algorithms 2-5, the proposed algorithm consists of multiple algorithmic blocks and introduces numerous hyperparameters—such as ρ, ν, δ, and p—that require tuning. These factors collectively detract from its practical value.

3. The numerical experiments are too trivial.

**Questions:**

1. As noted in the introduction, unconstrained consensus optimization is a special case of problem (1). For this special case, can the complexity of Algorithm 2 be reduced to match the optimal complexity of the unconstrained consensus algorithm in (Scaman et al., 2017)?

2. In the simulation section, how do you tune the hyperparameters?

3. The simulation results appear somewhat limited. It is difficult to observe whether the complexity scales proportionally with $\sqrt{\kappa_f}$, $\sqrt{\kappa_A}$, and $\sqrt{\kappa_W}$, as established by Theorem 1. It would be beneficial to empirically validate the complexity of the proposed algorithm, particularly with respect to $\kappa_f$, $\kappa_A$, and $\kappa_W$.

4. It is recommended to evaluate the algorithm using more real-world datasets and more complex experiments, such as logistic regression.

---

> ### Author Response · Authors · 2024-12-04
>
> Dear Reviewer,
>
> Thank you for your review.
>
> **Novelty, contribution and additional experiments**.
>
> For this part, please see the common answer to all Reviewers.
>
> **Can the complexity of Algorithm 2 be reduced to match the optimal complexity of the unconstrained consensus algorithm?**
>
> Thank you for the interesting question.
> $\newcommand{\mA}{{\mathbf A}}$
> As we see it, exact reduction is not possible in general.
> To reduce complexity bounds in Theorem 1 to that of Scaman (2017) we need to eliminate $\sqrt{\kappa_{\mathbf A}}$ from the communication complexity, and this is equivalent to $\kappa_{\mathbf A} = O(1)$.
> Since the coupled constraints must represent the consensus constraint $x_1 = \ldots = x_n \in \mathbb R^d$, the matrix $\mathbf A' := (\mathbf A_1 \ldots  \mathbf A_n)$ is required to have $\ker \mathbf A' = \mathcal L_d$, i.e., $\mathbf A' x = 0 \Leftrightarrow x_1 = \ldots = x_n$ for any $x = \text{col}(x_1, \ldots, x_n)$.
>
> For simplicity, let $d=1$, so all $\mA_i$ are column vectors. Natural examples of suitable matrices $\mathbf A'$ are incidence matrices of connected undirected graphs and their Laplacian matrices since $\ker \mA' = \mathcal L_1$. By the definition of the incidence matrix, the numerator of $\kappa_\mA$ is $L_\mA = \max_{i=1\ldots n}\sigma^2_{\max}(\mA_i) = \|\mA_i\|^2_2 = d_{\max}$, where $d_{\max}$ is the maximum degree of a vertex in the graph. Next, in the denominator we have $\mu_\mA = \frac1n\lambda_{\min^+}(\sum_{i=1}^n \mA_i \mA_i^\top)$. Note that $\sum_{i=1}^n \mA_i \mA_i^\top = \mA' \mA'^\top$, thus $\frac1n\lambda_{\min^+}(\sum_{i=1}^n \mA_i \mA_i^\top) = \frac1n\lambda_{\min^+}(\mA' \mA'^\top) = \frac1n\sigma_{\min^+}^2(\mA') = \frac1n\lambda_{\min^+}(\mA'^\top \mA') = \frac1n\lambda_{\min^+}(\mathbf L),$
> where $\mathbf L = \mA'^\top \mA$ is the Laplacian matrix of the same graph.
> Now, substituting this in the definition of $\kappa_\mA$, and using a common fact that $\lambda_{\max}(\mathbf L) \leq 2 d_{\max}$, we obtain $\kappa_\mA = \frac{n d_{\max}}{\lambda_{\min^+}(\mathbf L)} \geq \frac{n \lambda_{\max}(\mathbf L)}{2\lambda_{\min^+}(\mathbf L)} \geq \frac n2$. Similar calculations lead to the same $\kappa_\mA = \Omega(n)$ bound in the case then $\mA'$ is taken to be the Laplacian matrix of a connected graph.
>
> Therefore, using incidence matrix of any connected graph as $\mA'$  we have $\kappa_\mA = \Omega(n)$, what increases the number of communications by the factor of $\sqrt n$, comparing with the optimal convergence rates of Scaman. This is the additional price one need to pay for considering general coupled constraints instead of consensus constraints. This is typical for optimization: e.g., optimal algorithms for smooth (non-strongly) convex minimization have $O( 1/{k^2})$ convergence rates, while optimal algorithms for the more general class of smooth convex-concave saddle point problems only have $O(1/k)$ convergence when applied to minimization problems.
>
>
> **Parameter tuning**.
>
> The parameter values for all algorithms were chosen using the formulas from the papers to match the theoretically allowed ranges. Using the linear regression setup allowed us to calculate the parameter values analytically, so we did not use any black-box parameter tuning procedures

---

> ### Author Response · Authors · 2024-12-04
>
> **Dependence on condition numbers $\kappa_f, \kappa_{\mathbf A}, \kappa_{\mathbf W}$**.
>
> The dependence on $\kappa_{\mathbf A}$ and $\kappa_{\mathbf W}$ is quite clear from the theoretical analysis. When a matrix $\mathbf M$ is replaced with a Chebyshev polynomial of degree $\sqrt{\kappa_{\mathbf M}}$, the condition number of the resulting matrix becomes $O(1)$. This effectively removes its influence on the iteration complexity of the algorithm. Since the degree of the polynomial is explicitly defined, we know exactly how many matrix multiplications are required at each iteration.
>
> Conversely, the analysis of Nesterov's acceleration is notoriously non-intuitive, making it much harder to track the algorithm's complexity dependence with respect to $\kappa_f$. We conducted an additional experiment to validate that the number of gradient calls, $N$, is $O(\sqrt{\kappa_f})$. We used a consensus optimization linear regression setup similar to (22) with a ring graph on 5 nodes, $d=10$, and the Laplacian of the full graph on 5 nodes to represent the consensus constraints as coupled constraints. By varying $\kappa_f$ from $10^2$ to $10^6$, we counted the number of gradient calls required to achieve $\|x^k - x^*\|^2 \leq 10^{-5}$. We then used `scipy.stats.linregress` to estimate the parameter $\alpha$ in the dependence $N \approx c \kappa_f^\alpha$ using a log-log scale: $\log N \approx \log c + \alpha \log\kappa_f$. We obtained `LinregressResult(slope=0.496, intercept=2.018, rvalue=0.998, pvalue=3.343e-11, stderr=0.0105, intercept_stderr=0.098)`, where the slope corresponds to $\alpha$, and the intercept corresponds to $\log c$. The plot is available at https://ibb.co/jVJSQZZ. We observe that the estimated value of $\alpha = 0.496$ is very close to the theoretical value $\alpha=0.5$.

---

### Author Response · Authors · 2024-12-04
**Common answer to Reviewers**

Dear Reviewers,

We are grateful for your time, effort and the accurate reviews of our work. Thank you for acknowledging the novelty of the paper and paying attention to its strong parts. We upload a common answer to all of you and afterwards reply to each of the Reviewers individually. We hope that our replies will convince you to reconsider your scores.

The issues raised cover paper contribution and numerical experiments.

**Paper contribution**.

Reviewers **Rp6n** and **ieaZ** claim that the algorithm is a combination of existing methods and Reviewer **qaH7** asks to highlight the paper's contribution.

Indeed, we do use APAPC and (nested) Chebyshev acceleration together with the special problem reformulation (10), which include the decentralized-friendly reformulation of the coupled constraints and the augmented Lagrangian trick to induce the strong convexity. However, all these elements were not yet ready to be combined together: a nontrivial analysis was required to determine the strong convexity parameter of the reformulated objective on the necessary subspace (Lemma 1) and to derive the upper bound on the condition number of the matrix $B^TB$ (Lemma 2), as we need precise estimates for this quantities in terms of $\kappa_W$ and $\kappa_A$ (which are the initial assumptions) to derive the optimal convergence rates. One should also take into account that although the proof of Lemma 2 looks compact, it was tricky to obtain, especially because we did not know what the "correct" value of $\kappa_{\mathbf B}$ would be (for instance, what the "correct" form of Assumption 2 would be), as neither precise upper complexity bounds nor lower bounds were available before our work. In other words, it was impossible to obtain our main results using the analysis of previous works, and derivations of our results required new ideas.

Concluding, for the first time in the subfield of smooth and strongly convex distributed optimization with coupled constraints, we simultaneously develop both a state-of-the-art optimal algorithm and corresponding lower complexity bounds. This "resolves" this relatively small, yet important, subfield to a significant extent.  We thank you for raising a question on contribution. A corresponding clarification will be made in the revised version of the work.

**Numerical experiments**.

Reviewers **qaH7** and **HQPK** insist on including more experiments for different setups, including vertical federated learning (VFL) and decentralized optimization without coupled constraints, reviewer **Rp6n** says that the simulations are too trivial. We agree that the experiments in our paper are simple. However, we respectfully disagree that these experiments are *too* trivial. That is, the main purpose of these experiments is to demonstrate that the proposed theoretical results do not contradict our theory. The quadratic optimization problems serve this purpose very well because we have control over all the parameters of the problem, such as condition numbers. Our experiments align perfectly with the theory, demonstrating significantly improved convergence rates compared to the existing state of the art, as suggested by the theory.

We would also like to note that it is standard and common practice for strong theoretical papers published in top venues, such as ICLR, to have small *illustrative* experiments or not to have any experiments whatsoever. Application to more complex VFL scenarios is an interesting direction, however, we think it requires a separate study because of implementation details. At the same time, we compare our method to EXTRA (for consensus optimization) and find that our method outperforms EXTRA (we used linear regression setup similar to (22) with ring graph on 5 nodes, $d=10$, $\kappa_f = 10^4$ and Laplacian of the full graph on 5 nodes to represent the consensus constraints as coupled constraints). The plots can be found at https://ibb.co/gj1Tdck.

**Reduction to consensus optimization**.

The Reviewers are interested how our approach can be used for consensus optimization, i.e. decentralized optimization without additional affine constraints. Reviewer **Rp6n** is interested whether a theoretically optimal reduction may be done, while Reviewer **qaH7** asks for a numerical comparison with consensus optimization methods. In the personal answer to Reviewer **Rp6n**, we hypothesize that optimal complexity for decentralized optimization cannot be achieved directly from our approach. We do not see this as a problem of our method, since our algorithm is used for a more general problem class. We also carry out additional numerical experiments and verify that our method is competitive and even outperforms the decentralized optimization algorithm EXTRA proposed by Reviewer **qaH7** (see plots at https://ibb.co/gj1Tdck).

---

### Meta-Review · Area_Chair_vwoQ · 2024-12-17

**Metareview:**

The paper addresses decentralized optimization of a separable objective with affine coupled constraints, relevant to resource allocation, systems control, and distributed machine learning. It establishes lower complexity bounds and proposes a first-order algorithm that achieves these bounds, with linear convergence for general affine constraints—marking a first in this setting.

Most of the reviewers believe that the paper makes substantial contributions, and the AC also agrees with this assessment.

**Additional Comments On Reviewer Discussion:**

Four reviewers have evaluated the paper, and their overall assessment is positive. I agree with their evaluation and believe the paper offers a strong contribution with compelling results.

One reviewer believed that the contributions of the paper are limited and questioned the practical relevance of the paper. I believe the authors’ responses to these comments were satisfactory. Additionally, all reviewers raised a few technical questions, which the authors have addressed satisfactorily. I strongly recommend incorporating these remarks into the final version.

---

### Decision · Program_Chairs · 2025-01-22

Accept (Poster)